

# Toward a 3d Ising model with a weakly-coupled string theory dual

Nabil Iqbal[1] and John McGreevy[2]

**1** Centre for Particle Theory, Department of Mathematical Sciences,
Durham University, South Road, Durham DH1 3LE, UK
**2** Department of Physics, University of California at San Diego, La Jolla, CA 92093, USA

## Abstract

It has long been expected that the 3d Ising model can be thought of as a string theory, where one interprets the domain walls that separate up spins from down spins as two-dimensional string worldsheets. The usual Ising Hamiltonian measures the area of these domain walls. This theory has string coupling of unit magnitude. We add new local terms to the Ising Hamiltonian that further weight each spin configuration by a factor depending on the genus of the corresponding domain wall, resulting in a new 3d Ising model that has a tunable bare string coupling $g_s$. We use a combination of analytical and numerical methods to analyze the phase structure of this model as $g_s$ is varied. We study statistical properties of the topology of worldsheets and discuss the prospects of using this new deformation at weak string coupling to find a worldsheet description of the 3d Ising transition.

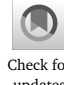
# 1   Introduction: Landau was right

Landau seems to have been even more right than we thought[1].

The Landau paradigm states that phases of matter (and the transitions between them) can be understood in terms of the symmetries that they spontaneously break [2]. Much recent work in many-body physics has focused precisely on phases and transitions that lie outside this paradigm; well-known examples are states distinguished by emergent gauge theory (topological order), which break no conventional symmetries and have no local order parameters [3–5].

Recently, however, we have learned to regard (many, if not all [6]) deconfined states of gauge theory in a new way: they spontaneously break *higher-form symmetries* [7]. Just as a conventional symmetry can lead to a conserved particle number, a higher-form symmetry enforces the conservation of a density of higher dimensional objects, such as strings or gauge flux tubes. Despite their slight unfamiliarity, these new symmetries do all the things that normal symmetries do: they can be continuous or discrete, they can spontaneously break (leading to Goldstone bosons in the continuous case [8,9]) and they can have anomalies. In many cases topological order can be shown to be essentially equivalent to such a (emergent, spontaneously broken) higher-form symmetry; of particular interest to us in this paper is the fact that the deconfined phase of Wegner's discrete 3d Ising lattice gauge theory spontaneously breaks such a $\mathbb{Z}_2$ one-form symmetry.

We are thus led to consider an *enlarged Landau paradigm*, one that includes both higher form symmetries and anomalies. These two additions to Landau's toolkit dramatically enlarge the set of systems that his paradigm describes; in addition to encompassing many examples of topological order, it seems that celebrated "beyond-Landau" phase transitions (such as the deconfined critical point between Neel and VBS phases in two dimensions [10]) can be understood in terms of (conventional) symmetries and their 't Hooft anomalies [11].

We turn now to a second tenet of the Landau paradigm: the critical point associated with a phase transition out of an ordered phase can be understood in terms of the fluctuations of the order parameter. In the case of spontaneous breaking of a one-form symmetry, the order parameter is a 'string field,' a field which creates excitations supported on loops. We can then ask ourselves whether the confinement transition of the 3d Ising gauge theory has a description in terms of the proliferation of strings. Note that the local data of this transition is the same as the 3d Ising transition (though it differs in global data and is sometimes called the Ising* universality class).

Indeed the existence of such a string theory description of the 3d Ising model has been proposed before [12, 13] (see references in [14]), with other motivations. In particular, one expects a close relation between the string worldsheet and the domain walls which separate regions of spin up and spin down. In the Ising gauge theory language, these surfaces are the sheets of flux. Indeed, following steps analogous to those which for the 2d Ising model produce the Jordan-Wigner solution in terms of free fermions, Polyakov argues for a fermionic stucture on the string

---

[1]This point of view has also been advocated recently in [1].

worldsheets, strongly suggestive of an RNS superstring [13].

An unsatisfactory aspect of this construction, in a proposed string theory reformulation of the nearest-neighbor cubic lattice 3d Ising model, was revealed by Distler [14]: the string coupling is not small. That is, the weight of a worldsheet of Euler character $\chi$ is proportional to $(-1)^\chi$, which says that $g_s = -1^2$.

In this paper, we propose to improve this situation by modifying the lattice Ising model in such a way as to make the dual string theory weakly coupled. That is, we change the microscopic Hamiltonian so that spin configurations whose domain walls have simple topology (smaller genus $g$, and hence larger $\chi = 2 - 2g$) have larger weight in the configuration sum:

$$Z = \sum_s g_s^{-\chi(s)} W_0(s),$$

where $W_0(s) = e^{-\beta \sum_{\langle ij \rangle} \sigma_i \sigma_j}$ is the usual Ising model Boltzmann weight. Here

$$\chi(s) \equiv N_F(s) - N_E(s) + N_V(s),$$

where $N_F, N_E, N_V$ are respectively the numbers of faces, edges and vertices of the dual lattice participating in a domain wall. The fact that the Euler character has this local representation makes it possible to implement this with short-range spin-spin interactions. Nevertheless, as we discuss in detail later, this statement requires some refinement.

We note that only when the correlation length is much bigger than the lattice spacing, $\xi \gg a$, *i.e.* near the phase transition, should we expect a description in terms of a *continuum* string theory. And of course the fixed-point values of the spin-spin interactions which describe the universal aspects of the critical behavior are *not* just nearest-neighbor interactions. Rather, all the couplings – including the coefficient of the Euler character $\phi \equiv \log g_s$ – will flow to some fixed-point value. That is, the critical Ising model has a fixed-point value of the string coupling $g_s$ which we cannot modify. Our proposal is to study a (non-universal) lattice model which initiates the flow in the weakly coupled regime, in hopes that this allows more physics of the weakly coupled string theory to reveal itself, on the way to the critical point.

Assuming that there is still a continuous phase transition in the modified model (there is), there are two alternatives: a) the modification has a critical point which represents a *new* universality class where proliferation of spherical domain walls dominates. The other possibility is b) the non-universal aspects of the transition, such as $T_c$, depend on $g_s$ but the universality class is the same. In our model we find both possibilities: for $g_s$ close to 1 we indeed remain in the usual Ising universality class, but for sufficiently small $g_s$ we encounter novel phases representing the proliferation of spherical worldsheets.

**Existing literature.** Quite a lot of work has been done in search of the string theory dual of the 3d Ising model [14, 16–28]. There are two points which give us hope for new progress. First, we are willing to modify the lattice model away from the nearest-neighbor model, which after all

---

[2]We note that a negative string coupling (in an unoriented string theory) plays a role in the recent paper [15].

is a non-universal demand. Second, quite a bit has been learned about non-perturbative string theory since the time of the work cited above. In particular, though most of our work is on the lattice, we will attempt to interpret this duality as holographic.

Ours is not the first study of random surfaces with Boltzmann weights more generic than the Ising model. Though our motivation is different, previous related work includes [29–35]. Here we make some comments on the literature.

The paper [32] studies random surfaces on the cubic lattice, attempting to keep track of their topology. The paper does not seem to address the issue of the ambiguities which we discuss below. In [36–38] it was observed that the genus of the surfaces making up the domain walls of the Ising model is generically nonzero, and the size distribution of handles was studied. They advocate a picture of coarse-grained interfaces made up of a distribution of microscopic handles. The paper [33] also considered a lattice discretization of random surfaces with weights which depend on their topology, a 'chemical potential' for the genus. The authors of [33] seem to be using a method similar to what we describe below as the "no-touching rule". They call the preferred phase at large $\chi$ a "droplet phase,"; it is closely related to what we describe as a "dilute phase". Finally, with soft-matter motivations, [34] give a mean-field treatment and map out a phase diagram including the phases we identify.[3]

**Organization.** The rest of the paper is organized as follows: after giving a general description of how the Boltzmann weight is to be modified, we identify a set of ambiguities in the prescription, arising from collisions of domain walls. We define two different versions of the model, which resolve the ambiguities in different ways. One is called the *no-touching model*, where the ambiguities are resolved by the simple expedient of disallowing all configurations which would have an ambiguity (their Boltzmann weight is zero). This has a cost, however, when simulating this model, since (in the high temperature phase at large density of domain walls) many updates will be rejected. The other resolution is called the *branch-point model*, where (following Distler [14]) we keep careful track of the connectivity of the domain walls, and include the contributions of *branch points* to the Euler character when required. In §3, we give a mean-field treatment of the phase diagram as a function of temperature and string coupling. We also describe some group theory on the cubic lattice and explain how to construct order parameters that diagnose the different phases we identify. In §4, we present the results of our numerical simulations, and in §5, we summarize our results and speculate about the nature of the worldsheet theory.

Many enjoyable details are relegated to appendices. In Appendix A we review the discrete

---

[3]We should comment on the apparently negative conclusions of [29–31]. They emphasize that the distribution of random surfaces they study is not dominated by smooth surfaces, but rather by very crumply fingery ones. Because of this they are pessimistic about the existence of a continuum limit. We must point out that this by itself does not problematize the existence of a useful dual string theory. For example, if we study free random walks, the ensemble of such walks is dominated by just such crumply fingery paths – objects with fractal dimension 2, very far from smooth curves (*e.g.* [39]). However, we can describe this perfectly well by a continuum field theory on the worldline. The same is true of the case of self-avoiding walks; though the average fractal dimension there is harder to compute, it is not 1, and yet we can use QFT, both on the worldline (the Edwards-Flory theory, *e.g.* [40, 41]) and in the target space (the Wilson-Fisher fixed point at $n = 0$ [42, 43]).

symmetries and defect operators of the 3d Ising spin model and lattice gauge theory using the modern language of higher form symmetries. In Appendix B we adapt the Wolff non-local update algorithm to both of our modified Ising models. In Appendix C we discuss some details of counting clusters of domain walls, and in Appendix D we present some details of the implementation of the Euler character terms in the Hamiltonian. Finally, full enjoyment of Appendix E requires little knowledge of statistical physics or string theory, but does require a pair of scissors and some tape.

## 2 Lattice construction

### 2.1 Usual 3d Ising model

We begin with a brief review of the usual Ising model on a 3d cubic lattice to establish notation. The partition sum of the Ising model may be written as follows:

$$Z = \sum \exp(-\widehat{H}), \qquad (2.1)$$

where the standard Ising Hamiltonian takes the form

$$\widehat{H} = \beta \sum_{\langle ij \rangle} \left(1 - \sigma_i \sigma_j\right), \qquad (2.2)$$

where $i, j$ run over the sites of the lattice, $\sigma_i = \pm 1$ denotes the spin on site $i$, and the sum $\langle ij \rangle$ runs over pairs of $i, j$ that are nearest neighbours. We have chosen to absorb a factor of the inverse temperature $\beta$ into the Hamiltonian for later convenience. We also introduce a notation where quantities with an overhat are "operators" in that they depend on the underlying classical spin configuration[4], unlike fixed parameters such as $\beta$.

Any spin configuration $\{\sigma_i\}$ on the original lattice can be thought of as specifying a configuration of domain walls on the dual lattice. A link between a site $i$ and its neighbouring site $j$ defines a face $\square$ of the dual lattice, and we say that this face hosts a domain wall if the spins at the two ends of the link disagree: $\sigma_i \neq \sigma_j$.

The factor $\left(1 - \sigma_i \sigma_j\right)$ appearing in the Ising Hamiltonian is nonzero only if a domain wall is present. The standard Ising Hamiltonian thus simply counts the (lattice) area $A$ of all domain walls:

$$H = 2\beta \sum_{\text{domain walls}} \widehat{A}. \qquad (2.3)$$

It is in this sense that the 3d Ising model is a kind of string theory, where (2.3) should be understood as a lattice-regularized version of the Nambu-Goto action. The inverse temperature $\beta$ is then the bare string tension in lattice units.

---

[4] We stress that we are doing classical statistical mechanics with $\sigma_i = \pm 1$, and there are thus no operators in the quantum-mechanical sense. (In this notation $\sigma_i$ should be more properly denoted $\widehat{\sigma}_i$, but as there is no scope for forgetting that $\sigma_i$ depends on the spin we have not done this.)

For future convenience, let us define the "wall operator"

$$\widehat{W}_{\langle ij \rangle} \equiv \frac{1}{2}(1 - \sigma_i \sigma_j),\qquad(2.4)$$

which is defined on a link of the original lattice, or equivalently on a face $\square$ of the dual lattice. It is 1 if a domain wall is present on this face, and is 0 otherwise.

The usual Ising model has two phases. As $\beta \to \infty$ all fluctuations are suppressed, and the energy is clearly minimized if we forbid all domain walls. We thus find an ordered phase where all spins are aligned. This ferromagnetic phase spontaneously breaks the spin symmetry, and there is a net magnetization: $\langle \sigma \rangle \neq 0$. For small $\beta \ll 1$, fluctuations are no longer suppressed and we find a disordered paramagnetic phase with $\langle \sigma \rangle = 0$. These phases are separated by the usual 3d Ising transition; on the cubic lattice this at $\beta = \beta_c \approx 0.221$ (see e.g. [44] for a recent high-precision determination of this critical coupling). The Ising CFT is strongly coupled with no obvious small parameters; however a great deal is known about it, and the current most precise determination of the critical exponents arises from the conformal bootstrap [45] (see [46] for a review).

If we allow $\beta$ to be negative, there is a further ordered *antiferromagnetic* phase at large negative $\beta$, in which spins are anti-aligned, $\langle \sigma_{xyz} \rangle \sim (-1)^{x+y+z}$. In fact the 3d Ising model is mapped to itself under the map $\sigma_{xyz} \to (-1)^{x+y+z} \sigma_{xyz}, \beta \to -\beta$. Thus, in the unmodified Ising model, there is no new dynamical information at negative $\beta$: the paramagnetic/antiferromagnetic transition is at precisely $\beta = -\beta_c$ and is equivalent (up to a field redefinition) to the ferromagnetic transition.

## 2.2 Euler character of lattice domain walls

In the conventional formulation of worldsheet string theory, one sums over all embeddings of a string worldsheet in an appropriately defined target space. The weight of each configuration to the partition sum is given by the Nambu-Goto action, plus a topological term:

$$S = \int d^2x \sqrt{h}\left(\frac{1}{2\pi\alpha'} + \frac{1}{4\pi}\phi R(x)\right),\qquad(2.5)$$

where $h$ is the induced metric on the worldsheet, $\phi$ is the dilaton, and $R$ is the 2d Ricci scalar. The first term simply measures the area of the worldsheet, and so is analogous to the usual 3d Ising Hamiltonian (2.3), where we should identify the inverse temperature $\beta$ with the string tension $(2\pi\alpha')^{-1}$. However the second term measures the Euler character $\chi$ of the string worldsheet, weighting each contribution by a factor of $g_s^\chi$, where the string coupling $g_s = e^\phi$. This dependence on the Euler character is crucial in the perturbative formulation of string theory, where worldsheets with more handles are suppressed at weak coupling. Such a term is not present in the usual Ising model.

We now seek to add a term measuring this Euler character to the Ising Hamiltonian. As we explain below, this can be done in a local manner, resulting in a decorated but still local version

of the 3d Ising Hamiltonian that has two tunable parameters: the string tension $\beta$ and the string coupling $g_s$.

How do we find the Euler character of a lattice surface $s$? If $s$ has $N_F$ faces, $N_E$ edges, and $N_V$ vertices, then we simply compute

$$\chi \equiv N_F - N_E + N_V \, . \tag{2.6}$$

We thus need to extract these quantities from the spin data above. We note that all of these geometric quantities naturally live on the dual lattice; thus in the remainder of this section we will refer only to faces, edges and vertices of the dual lattice.

### 2.2.1 Faces

Computing the number of faces $N_F$ is very simple: as explained above, this is precisely what the usual Ising Hamiltonian does, and we can conveniently write the answer in terms of the wall operator:

$$\widehat{N}_F[s] = \sum_{\text{faces}} \widehat{W}_\square \, , \tag{2.7}$$

where the sum runs over all faces of the dual lattice.

### 2.2.2 Edges I

Computing the number of edges is only slightly more complicated. A given edge of the dual lattice has four faces incident on it, and there are eight possible configurations $\mathcal{E}$ of domain walls populating these faces, shown in Figure 1. Depending on which configuration we have, a given edge should contribute to the sum in (2.6) as being part of either zero, one, or two (touching) domain walls. We will denote this number by $D_{\mathcal{E}} \in 0, 1, 2$.

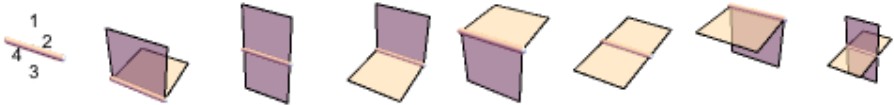

Figure 1: The configurations near an edge consistent with the fact that a domain wall has no boundaries.

We now note that through judicious use of the wall operator, we can construct a projector onto each of these configurations. For example, if we denote the four faces as $\square_m$, $m \in 1 \cdots 4$ then a projector that is 1 for the second configuration from the left in Figure 1 and zero for all others is

$$\widehat{P}_{\mathcal{E}} = \widehat{W}_{\square_1} \widehat{W}_{\square_2} (1 - \widehat{W}_{\square_3})(1 - \widehat{W}_{\square_4}) \, . \tag{2.8}$$

We can now use these projectors to add a combination of local terms to the Hamiltonian that assign each of the six possible configurations any desired weight, defining an operator $E_-$ defined

on each edge $-$ of the dual lattice.

$$\widehat{E}_- = \sum_{\mathcal{E}} D_{\mathcal{E}} \widehat{P}_{\mathcal{E}}. \tag{2.9}$$

Summing this operator over every edge of the dual lattice we will count the total number of edges.

$$\widehat{N}_E[s] = \sum_{\text{edges}} \widehat{E}_-. \tag{2.10}$$

We note that the last case – that in which there are four incident domain walls on the edge, and the edge is counted as 2 – is different from the rest, in that one has to make a choice on how to interpret the surface represented by the lattice data – see. e.g. Figure 2. All of these choices however result in the edge contributing 2 to the final sum, so this ambiguity does not affect our final Hamiltonian. We will see that for vertices the situation is rather more complicated.

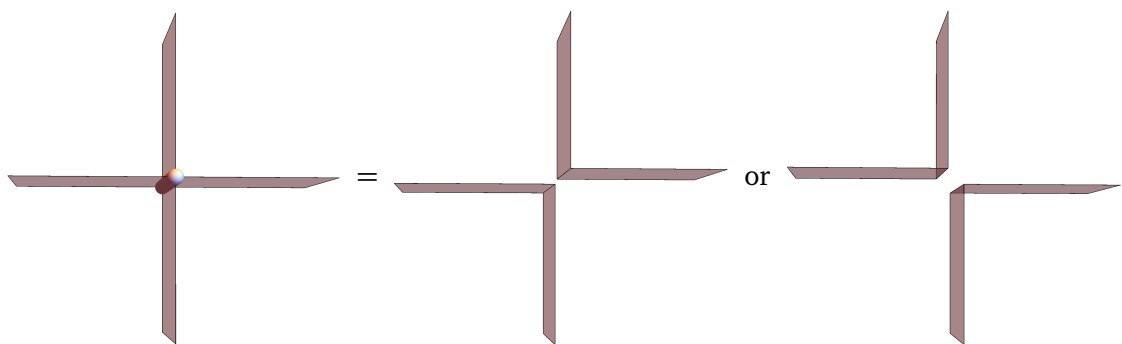

Figure 2:   Two ways of resolving an edge with four incident walls.  There is yet another choice (not shown) where it is resolved into two intersecting straight domain walls.

### 2.2.3   Vertices

Vertices are conceptually similar, but practically somewhat more difficult. Given a configuration, such as Figure 3, of faces neighbouring a given vertex of the dual lattice, we would like to determine its contribution to the Euler character. As in the case of the edges discussed above, this is equivalent to determining a prescription to separate the wall configuration specified by the lattice into distinct closed (possibly intersecting) surfaces, and then counting the number of surfaces in the decomposition.

We note that a given vertex of the dual lattice is surrounded by 8 spins. An overall flip of all the spins leaves the configuration of domain walls invariant, and so there are $2^{8-1} = 128$ different possibilities, corresponding to the number of ways to populate the 12 incident domain walls with a closed surface. Each of these configurations determines a string of 12 bits $\mathcal{V} = \{n_{\square_i}\}$, where

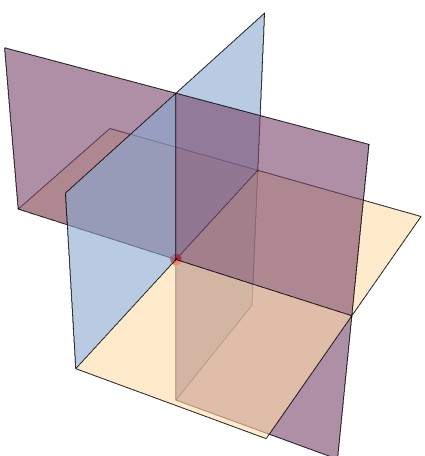

Figure 3: An example of a possible configuration of domain walls around a vertex.

for each $i \in 1 \cdots 12$, $n_{\square_i}$ is either 1 or 0 depending on whether a domain wall is present on the corresponding face or not.

Just as in (2.8), we can define a projection operator onto each of these configurations:

$$\widehat{P}_\mathcal{V} = \prod_{i=1}^{12} (\widehat{W}_{\square_i})^{n_{\square_i}} (1 - \widehat{W}_{\square_i})^{1-n_{\square_i}} . \tag{2.11}$$

To write down a term in the Hamiltonian, we must now assign a vertex number $D_\mathcal{V}$ – corresponding to the number of surfaces in the decomposition – to each of these 128 configurations. For many choices of wall configuration this is intuitively obvious. However as we increase the number of walls we come to an interesting issue. Just as for the edge configuration in Figure 2, it turns out that many of the configurations can be separated into distinct domain walls in more than one way. *Unlike* the edge configuration, some of these distinct choices result in different *numbers* of domain walls in the separation (see Figure 4). Thus, just to assign a vertex number to the configuration, we must make a decision on how to resolve this ambiguity.

In this work, we study two different models which resolve the ambiguity in distinct ways:

1. *No-Touching Model:* This is the simplest choice. Here we simply enumerate all possible vertex choices that have a possible ambiguity (i.e. those which have an edge with four edges incident on it, as in the right-most case in Figure 1) and assign them each an extremely high energy cost, forbidding them from contributing to the partition sum. Physically this corresponds to a short-ranged interaction that prohibits collision of domain walls. Though

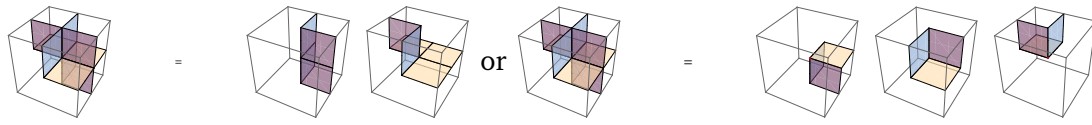

Figure 4: The contribution of a given vertex to the euler character is ambiguous. In this example, depending on the chosen resolution, it is either 2 or 3.

conceptually simple, there are two concrete disadvantages to this approach: the conventional 3d Ising model is no longer a precise limit of this model, and simulations using this model are very slow deep in the disordered phase (where there are many domain walls and this constraint plays a role.)

2. *Branch-Point Model:* Here we make a more sophisticated choice: for every possible vertex configuration (save one, explained in Appendix D, where symmetry dictates a different choice), we choose to decompose the vertex in the way that results in the *maximum* number of domain walls. This fixes the vertex number, and assigns an Euler character to every distinct spin configuration. This has the benefit that the usual 3d Ising model is obtained after setting $g_s = 1$. There is however a new complication that must be dealt with, discussed in the next subsection.

In either case, we now obtain a vertex number $D_\mathcal{V} \in \{1, \cdots 4\}$ for each of the 128 different configurations. The details of this computation are outlined in Appendix D. We then construct a vertex-counting operator, defined on each vertex $\odot$ of the dual lattice

$$\widehat{V}_\odot = \sum_\mathcal{V} D_\mathcal{V} \widehat{P}_\mathcal{V} \,. \tag{2.12}$$

Summing this over each vertex of the dual lattice we obtain the total vertex number:

$$\widehat{N}_V[s] = \sum_{\text{vertices}} \widehat{V}_\odot \,. \tag{2.13}$$

Finally, we observe that all these ambiguities may be avoided (as in [47]) by studying the Ising model on a lattice where each edge has exactly three faces incident upon it (rather than four as for the cubic lattice). An example is the cuboctohedron lattice. We leave a study of this lattice to future work.

### 2.2.4 Edges II: branch points

In the case of the *branch-point model* discussed above, we now run into a new issue. A choice of how to separate and connect the domain walls is made at each vertex. It is now possible that the choices made at two neighbouring vertices, joined by an edge, will not be compatible, as shown in Figure 5. This compromises the interpretation of the lattice data as a configuration of closed

surfaces, e.g. by sometimes resulting in *odd* Euler characters, which is never possible for a closed oriented surface[5].

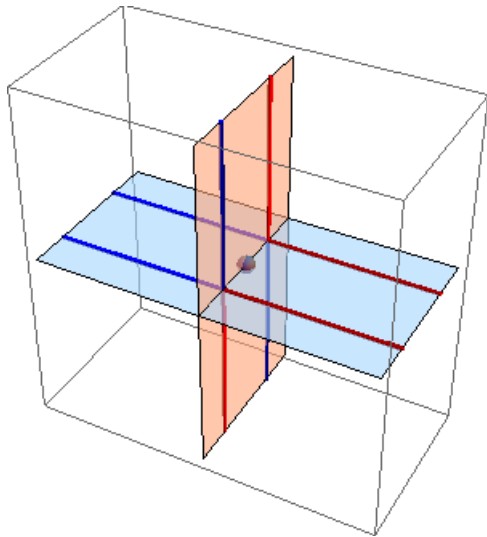

Figure 5: The vertices at the two ends of this edge have been decomposed in incompatible ways. When each vertex is pulled apart into different topologically distinct surfaces by picking one of the options in e.g. Figure 4, the individual faces emanating from the vertex are joined pairwise in a particular manner. Here the red lines both belong to one such surface and the blue lines to another. The resulting incompatibility requires us to add a branch point (grey dot) along the edge to rejoin the faces.

A possible disagreement will happen when the edge has four domain walls incident on it, the rightmost case in Figure. 1, with a naive edge number of 2. It can be rectified by noting that it is possible to introduce a *branch point* along the edge to reconnect the vertices, as in Figure 5. This corresponds to adding two new edges but only *one* new vertex, meaning that the contribution of the branch point to the Euler character is

$$\Delta\chi_{\text{branch}} = -1. \tag{2.14}$$

Another way to arrive at the same result is to note that this is a curvature singularity. In all cases, circling around the singularity we make a full circle around both of the two surfaces, resulting in a total opening angle of $4\pi$. (The reader may now find it helpful to pause and assemble the paper cutout provided in Appendix E). The contribution of a curvature singularity of opening angle $\alpha$ to the Euler character is

$$2\pi\Delta\chi_{\text{branch}} = (2\pi - \alpha), \tag{2.15}$$

resulting in the same expression as above.

---

[5]In general, domain walls may form a triple point, where three sheets of the immersion meet transversally. For smooth immersions, the number of such triple points equals the euler character mod two [48].

Comparing with (2.6), we summarize this by stating that the edge containing a branch point contributes to the Euler character with a total weight of 3 (rather than 2). To implement this in our Ising Hamiltonian, we need to modify our edge-counting operator (2.9). It is straightforward to enumerate the possible decompositions and write down a (cumbersome) series of projectors $P^{\mathrm{b}}_{\mathcal{V}_1,\mathcal{V}_2}$ onto the two vertices $\odot_1$, $\odot_2$ neighbouring the edge that adds 1 to the edge weight when the decompositions on the two sides do not match:

$$\widehat{E}_- = \sum_{\mathcal{E}} D_{\mathcal{E}} \widehat{P}_{\mathcal{E}} + \sum_{\mathcal{V}_1 \in \odot_1} \sum_{\mathcal{V}_2 \in \odot_2} \widehat{P}^{\mathrm{b}}_{\mathcal{V}_1,\mathcal{V}_2}. \tag{2.16}$$

The details of how this is implemented are explained in Appendix D.

### 2.3 3d Ising string theory

We can now finally write down the *3d Ising string theory* Hamiltonian:

$$\widehat{H} = 2\beta \sum_{\text{faces}} \widehat{W}_{\square} + \phi \left( \sum_{\text{faces}} \widehat{W}_{\square} - \sum_{\text{edges}} \widehat{E}_- + \sum_{\text{vertices}} \widehat{V}_{\odot} \right). \tag{2.17}$$

This Hamiltonian is a function of two parameters, the string tension $\beta$ and the dilaton $\phi$. Each spin configuration can be understood as a collection of closed string worldsheets with area $A$ and Euler character $\chi$, which contribute to the partition sum as $e^{-2\beta A - \phi \chi}$. Note that the string coupling $g_s = e^{\phi}$ is now an ordinary tunable parameter.

The addition of a tunable string coupling $g_s$ encourages us to think of this model as a non-perturbatively defined lattice string theory. We stress that this is a perfectly ordinary spin Hamiltonian with a certain carefully chosen pattern of next-to-next-to-next-to-nearest neighbour interactions. It may thus be attacked using conventional statistical-mechanics techniques.

## 3 Expectations

In this section we present a mean-field treatment of the Hamiltonian above, discussing the expected phase structure in the $(\beta, \phi)$ plane.

We assume throughout this section that the system can be understood in terms of a unit cell of size $2 \times 2 \times 2$. As the system has interactions that couple spins to neighbouring spins that are 3 sites away, this is an assumption, though one that seems to be borne out by our numerics.

### 3.1 Possible ordered phases

The Hamiltonian (2.17) takes the form

$$H = 2\beta A + \phi \chi. \tag{3.1}$$

At large positive (negative) values of $\beta$ and $\phi$, we expect to find ordered phases that minimize (maximize) the worldsheet area $A$ and Euler character $\chi$ respectively.

We assume a $2 \times 2 \times 2$ unit cell. It is straightforward to determine the spin configurations that extremize $A$ and $\chi$ per unit cell [6]. Below, we explicitly show the pattern of eight spins in the unit cell (by showing two slices through the cube) for each configuration. The resulting extremal configurations are slightly different depending on which of the two vertex resolution protocols we use.

### 3.1.1 Branch point vertex resolution

The possible vacuum configurations are:

1. Minimum $A$: *ferromagnetic phase* ($A = 0, \chi = 0$). This is the usual ferromagnetic ground state with all spins aligned; it has no domain walls, and thus clearly has vanishing area and Euler character.



2. Maximum $A$: *anti-ferromagnetic phase* ($A = 24, \chi = 0$). This is the usual anti-ferromagnetic ground state, where all spins are anti-aligned with their nearest neighbours: $\sigma_{x,y,z} = (-1)^{x+y+z}$. This clearly maximizes the number of domain walls; every possible face is occupied by a domain wall. It has Euler character of 0, as the ambiguitiy resolution protocol described above resolves this as a series of three non-intersecting perpendicular planes, each with toroidal topology and thus vanishing Euler character.

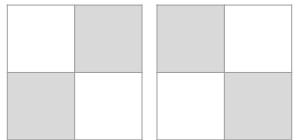

3. Maximum $\chi$: *packed* phase ($A = 18, \chi = 6$). This is obtained from the anti-ferromagnetic phase by flipping a single spin (as all spins are equivalent up to a symmetry operation, it does not matter which spin we flip). It can be understood as three spherical domain walls that are packed into the unit cell as closely together as possible given the vertex resolution protocol described in the previous section. We show the packed phase explicitly in Figure 6.

---

[6]Equivalently, we assume the full system is a $2 \times 2 \times 2$ torus with periodic boundary conditions, and quote the value of $A$ and $\chi$ for this system.

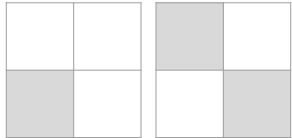

4. Minimum $\chi$: There are two degenerate configurations here with precisely the same values for $A$ and $\chi$: $(A = 12, \chi = -4)$.

   The first is the *plumber's nightmare I,* where we follow the nomenclature of [34], who identify a similar phase in a continuum model. This can be visualized as three small tubes that open into a small "room" in the middle. When the unit cell is replicated, this results in a geometry where the genus of the final surface is negative, growing extensively with the volume.

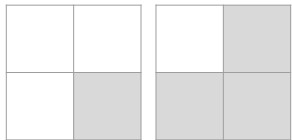

   The second is the *plumber's nightmare II*, which appears superficially similar as a network of tubes.

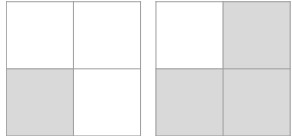

   As we will discuss in the next section, these two similar-looking configurations actually break rather different patterns of lattice symmetries.

### 3.1.2 No-touching vertex resolution

The no-touching rules explained in Section 2.2.3 forbid some of the above configurations. We thus find slightly different results:

1. Minimum $A$: *ferromagnetic phase* $(A = 0, \chi = 0)$, as for the branch-point protocol.

2. Maximum $A$: The previous antiferromagnetic phase is excluded by the no-touching rules. Interestingly, we now find a degeneracy between the two *plumber's nightmares I* and *II*, $(A = 12, \chi = -4)$ and a new phase which we call the *dilute* phase $(A = 12, \chi = 4)$, with spin configuration shown below:

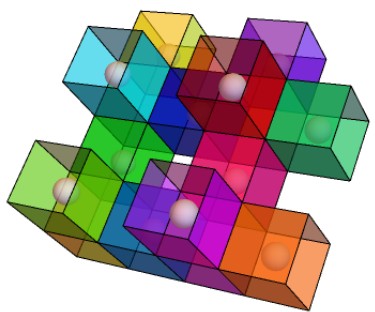

Figure 6: Example of the packed phase on a $2 \times 4 \times 4$ lattice; spheres represent spins that are up, and empty spaces denote spins that are down. Domain walls are explicitly shown, and different clusters have been given different (randomly chosen) colors.

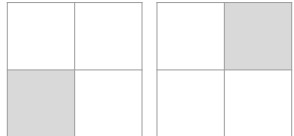

The unit cell consists of two spherical domain walls, rather than three as in the packed phase. As the dilute phase has a positive $\chi$, it will be preferred over the plumber's nightmares at large positive $\phi$.

3. Maximum $\chi$: *dilute* phase ($A = 12, \chi = 4$). The packed phase is now excluded by the no-touching protocol.

4. Minimum $\chi$: We have a degeneracy between the *plumber's nightmares I* and *II* ($A = 12, \chi = -4$) as before.

Having understood these classical phases, one can ask what configuration minimizes the Hamiltonian (3.1) for given values of $(\beta, \phi)$. As we take $\beta, \phi \to \pm\infty$, fluctuations are suppressed and we expect the system to be in an ordered phase close to one of the "vacuum" states described above. In this way one obtains the "mean-field" phase diagrams shown in Figure 7. Note that the two plumber's nightmare phases have identical "charges" (i.e. $A$ and $\chi$), and so the choice between them cannot be made from mean-field theory considerations. (Interestingly, below we observe numerically that the two protocols for resolving vertices make different choices between these two phases; we do not have a simple argument as to why.) These mean-field phase diagrams will also clearly will not capture the physics at small $\beta, \phi$, where we expect fluctuations and the existence of a disordered phase.

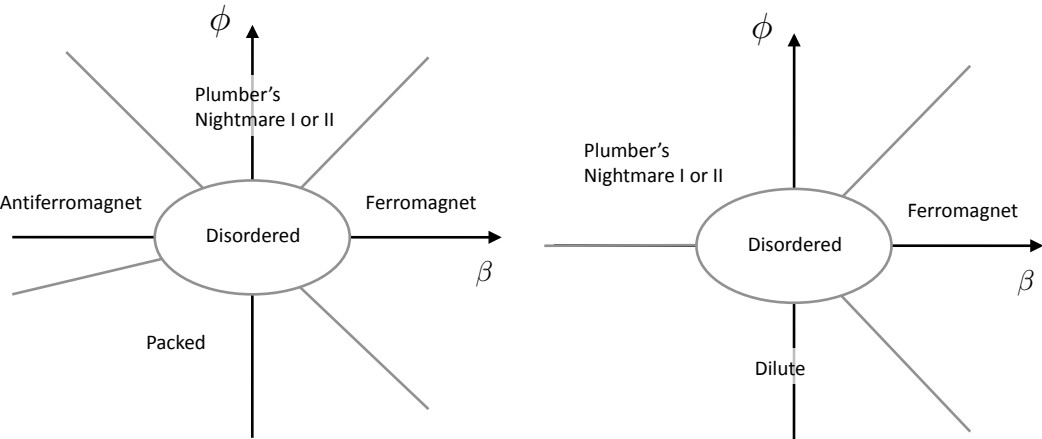

Figure 7: "Mean-field" phase diagrams (arising from minimizing classical Hamiltonian) for branch-point vertex resolution (*left*) and no-touching vertex resolution (*right*). Note that as the two plumber's nightmare phases have the same $A$ and $\chi$ this energetic analysis cannot distinguish between them. At small $\beta, \phi$ we expect to find a disordered phase.

## 3.2 Order parameters

We note that while we have presented configurations that minimize the energy, it is perhaps not entirely obvious that each of these novel "vacuum" configurations – e.g. in particular the "packed" and "plumber's nightmare" – necessarily constitute different *phases* at finite temperature. One way to guarantee that they are distinct phases is if they each result in a distinct pattern of breaking of the global symmetries of the underlying spin Hamiltonian. As we will show, this is true for most (but not all) of the phases.

More precisely, given a microscopic global symmetry group $G$, we say that this is broken down to a subgroup $H \subset G$ if there exists a multiplet of operators $\widehat{O}_i$ such that $G$ acts linearly on $\widehat{O}_i$, $H$ leaves $\widehat{O}_i$ invariant, and $\langle \widehat{O}_i \rangle \neq 0$.

We now discuss in detail how to understand this pattern of symmetry breaking in our case, as well as how to use it to construct order parameters that allow us to identify each of the above phases in the numerical simulations. It is clear that the $O_i$ in our case are linear combinations of spins. With an eye towards later generalizations, we take a somewhat abstract point of view to describe *which* linear combinations are of interest.

### 3.2.1 Global symmetries

We first discuss the global symmetries of our system. There is a global $\mathbb{Z}_2$ spin flip symmetry

$$\mathbb{Z}_2 : \sigma_{x,y,z} \rightarrow -\sigma_{x,y,z} . \tag{3.2}$$

There are also lattice symmetries. We imagine periodic boundary conditions on a torus of length $N$; in that case the system has a rather large symmetry group, allowing translations by $N$ sites. We assume that this large symmetry is dynamically broken *at most* to the symmetries of a $2 \times 2 \times 2$ unit cell[7]. Then the symmetry group of our microscopic Hamiltonian can be taken to be:

$$G = S_4 \times (\mathbb{Z}_2)^3 \times \mathbb{Z}_2, \tag{3.3}$$

where $S_4$, the permutation group on four elements, is the (orientation-preserving) symmetry group of rotations of the cube, $(\mathbb{Z}_2)^3$ corresponds to translating the unit cell by one site in each of the three directions, and the final $\mathbb{Z}_2$ is the overall spin symmetry (3.2).

We would now like to understand how these symmetry operations act on the spins. Let us denote the eight spins in the unit cell by $\sigma_{x,y,z}$ with $x, y, z \in \{0, 1\}$. We construct an arbitrary linear combination of these spins $s$ as

$$s = \sum_{x,y,z \in \{0,1\}} s^{x,y,z} \sigma_{x,y,z}. \tag{3.4}$$

It will sometimes be convenient to arrange the eight coefficients $s^{x,y,z}$ into a vector $s^I$ as

$$s^I = \left( s^{000}, s^{100}, s^{010}, s^{110}, s^{001}, s^{101}, s^{011}, s^{111} \right). \tag{3.5}$$

The elements of $G$ have a linear action on this eight-element vector, found by demanding that the abstract object $s$ remains invariant under simultaneous transformation of $s^{x,y,z}$ and $\sigma_{x,y,z}$. Lattice symmetries permute the spins amongst each other. For example if we translate by $x$:

$$T_x s^{x,y,z} = s^{x+1,y,z} \tag{3.6}$$

and similarly for $y$ and $z$, where the addition in the unit cell is done modulo 2. Note that as far as the unit cell is concerned, the translation in the $x$ direction is equivalent to a reflection in the $yz$ plane.

To understand the action of $S_4$, we note that the eight-component object $s$ breaks into the following direct sum of irreducible representations under $S_4$:

$$\text{FM}_1 \oplus \text{AFM}_1 \oplus O_3^+ \oplus O_3^-. \tag{3.7}$$

Here we have used a notation appropriate to our problem, denoting the dimension of the representation by the subscript.

1. FM is the one dimensional trivial rep

$$s^I_{\text{FM}} = (1, 1, 1, 1, 1, 1, 1, 1), \tag{3.8}$$

   left invariant by $S_4$ (and, indeed by all lattice symmetries). It corresponds to taking a linear combination of all the spins with uniform weight, and so is a ferromagnetic order parameter.

---

[7]In other words, we assume the phases of the system can be classified by studying the symmetries of the length-2 unit cell, and assume that the dynamics does not realize the further breaking necessary to obtain a (e.g.) $4 \times 4 \times 4$ unit cell.

2. AFM is the one dimensional sign rep

$$s^I_{\text{AFM}} = (-1, 1, 1, -1, 1, -1, -1, 1) \, . \tag{3.9}$$

Under $S_4$ it transforms by a factor of the sign of the permutation. From our point of view, it is an alternating sum over spins and measures antiferromagnetic order. This is invariant under all lattice symmetries, provided they are appropriately combined with a global spin flips.

3. $O^+$ is a 3-dimensional rep; from the point of view of $S_4$, it is the so-called "standard representation". Explicitly, it is spanned by the following orthonormal basis

$$s^I_{+,i} = \begin{pmatrix} \frac{1}{2} & 0 & 0 & -\frac{1}{2} & -\frac{1}{2} & 0 & 0 & \frac{1}{2} \\ -\frac{1}{2\sqrt{3}} & \frac{1}{\sqrt{3}} & 0 & -\frac{1}{2\sqrt{3}} & -\frac{1}{2\sqrt{3}} & 0 & \frac{1}{\sqrt{3}} & -\frac{1}{2\sqrt{3}} \\ -\frac{1}{2\sqrt{6}} & -\frac{1}{2\sqrt{6}} & \sqrt{\frac{3}{8}} & -\frac{1}{2\sqrt{6}} & -\frac{1}{2\sqrt{6}} & \sqrt{\frac{3}{8}} & -\frac{1}{2\sqrt{6}} & -\frac{1}{2\sqrt{6}} \end{pmatrix}, \tag{3.10}$$

where here $i \in 1, 2, 3$ runs over these three basis vectors. On this subspace, we have $T_x T_y T_z = \mathbf{1}$; however it transforms under all elements of $S^4$ and under global spin flips.

4. $O^-$ is another 3-dimensional rep, explicitly spanned by

$$s^I_{-,i} = \begin{pmatrix} -\frac{1}{2} & 0 & 0 & \frac{1}{2} & -\frac{1}{2} & 0 & 0 & \frac{1}{2} \\ -\frac{1}{2\sqrt{3}} & -\frac{1}{\sqrt{3}} & 0 & -\frac{1}{2\sqrt{3}} & \frac{1}{2\sqrt{3}} & 0 & \frac{1}{\sqrt{3}} & \frac{1}{2\sqrt{3}} \\ -\frac{1}{2\sqrt{6}} & \frac{1}{2\sqrt{6}} & -\sqrt{\frac{3}{8}} & -\frac{1}{2\sqrt{6}} & \frac{1}{2\sqrt{6}} & \sqrt{\frac{3}{8}} & -\frac{1}{2\sqrt{6}} & \frac{1}{2\sqrt{6}} \end{pmatrix}, \tag{3.11}$$

This rep can be distinguished from $O^+$ by noting on this subspace we have $T_x T_y T_z = -\mathbf{1}$. It can be obtained by tensoring $O^+$ with the one dimensional AFM representation in (3.9). It transforms under all elements of $S_4$, but is invariant under global spin flips if they are combined with $T_x T_y T_z$.

### 3.2.2 Constructing order parameters

Now, for each choice of representation $r$ from the list of four above, we can construct an order parameter as follows. Given a microscopic spin configuration $\sigma_{x,y,z}$, we compute the overlap of the spin data with each basis element $i \in r$ of the appropriate invariant subspace by taking an inner product as:

$$\widehat{O}_{r,i} = \frac{1}{N^3} \sum_{x,y,z=1}^{N} s^{[x,y,z]}_{r,i} \sigma_{x,y,z} \tag{3.12}$$

(where the notation $[x, y, z]$ means that all elements are taken modulo 2 to project down into the unit cell).

These $\widehat{O}_{r,i}$ are the desired order parameter operators. For $r = \{\text{FM}, \text{AFM}\}$, the $i$ index has no meaning as the order parameters transform in 1-dimensional reps under the rotation group. For $r = O^\pm$ the $i$ index transforms in the 3-dimensional reps described above. A nonzero value

Table 1: Order parameters (columns) which are nonzero in various vacuum configurations (rows), together with symmetries preserved by each phase. Note that the Plumber's nightmare II and the packed phase break the same symmetries and thus belong to the same phase.

| **Phase** | FM | AFM | $O^+$ | $O^-$ | Unbroken subgroup $H \subset G$ |
|---|---|---|---|---|---|
| Paramagnetic | 0 | 0 | 0 | 0 | All |
| FM | × | 0 | 0 | 0 | Lattice symmetries $S_4 \times (\mathbb{Z}_2)^3$ |
| AFM | 0 | × | 0 | 0 | Lattice symmetries $S_4 \times (\mathbb{Z}_2)^3$ combined with spin flip $\mathbb{Z}_2$ |
| Plumber's nightmare I | 0 | × | 0 | × | $T_x T_y T_z$ combined with spin flip $\mathbb{Z}_2$ |
| Plumber's nightmare II | 0 | 0 | × | × | None |
| Packed | × | × | × | × | None |
| Dilute | × | 0 | × | 0 | $T_x T_y T_z$. |

for any of these four order parameters implies a particular pattern of symmetry breaking, and so defines a phase. We have described above which subgroups of $G$ are left invariant by each order parameter.

For $r = O^{\pm}$, it is convenient to define a fully invariant scalar order parameter by taking the norm:

$$\widehat{\mathcal{O}}_r = \sum_{i=1}^{3} \widehat{O}_{r,i}^2 \tag{3.13}$$

(This norm is invariant under $G$ given the choice of normalized basis vectors (3.10) (3.11)). It is straightforward to check which of these order parameters are nonzero in the vacuum states described above; the results are shown in Table 1.

We see that most of the vacuum configurations described above correspond to different unbroken subgroups $H \subset G$, and thus to distinct phases. An exception is associated with the configuration named Plumber's Nightmare II: this is energetically degenerate with the Plumber's Nightmare I, but breaks rather different symmetries – in fact, from the pattern of symmetry breaking, it is equivalent to the Packed phase. This peculiar state of affairs probably arises from our fine-tuning of the Hamiltonian to be sensitive only to $A$ and $\chi$. Note that in principle any combination of the order parameters could be nonzero, though it appears not all possibilities are dynamically realized.

This discussion has been rather abstract. To make it concrete, let us explicitly evaluate the case $r =$ FM and $r =$ AFM. Putting (3.8) and (3.9) into (3.12), we find

$$\widehat{O}_{\text{FM}} = \frac{1}{N^3} \sum_{x,y,z} \sigma_{x,y,z} \qquad \widehat{O}_{\text{AFM}} = \frac{1}{N^3} \sum_{x,y,z} (-1)^{x+y+z} \sigma_{x,y,z} \tag{3.14}$$

i.e. we are simply computing the mean magnetization and staggered magnetization, as expected.

Finally, we note that we can use this information to understand the transitions between different phases. We expect a continuous transition between two phases only when a *single* order

parameter condenses. (A continuous, direct transition with different order parameters on the two sides [10] would be an extremely interesting surprise, but generally requires an intimate, nematic relation between the two order parameters.) Thus a continuous transition is possible between the paramagnetic and FM or AFM phases (this is the usual Ising transition), and between the AFM and plumber's nightmare phase (i.e. the condensation of the $O^-$ order parameter alone; to the best of our knowledge, this is a novel universality class). All other transitions are expected to be first order.

**Expectations for critical behavior.** Consider the conformal field theory governing the 3d Ising critical point. It has a single relevant $\mathbb{Z}_2$-invariant operator. The $\mathbb{Z}_2$-invariant perturbation we are making of the lattice model has some overlap with this relevant operator, since from (2.6), $\chi = $ area + other terms. So we generically expect nonzero $\phi$ to shift the critical temperature. But, since there are no *other* $\mathbb{Z}_2$-invariant relevant operators in the Ising CFT, this shift of the critical temperature should be the only effect on the critical behavior at small $|\phi|$.

An analogous argument was made in [49] in the case of self-avoiding walks perturbed by a chemical potential for *writhe*, a topological invariant which also maps to a local perturbation of the field theory description.

# 4 Numerical results

In this section we present the results from a preliminary Monte Carlo investigation of the model described above. Our results may be summarized as follows:

1. We find that the expectations from the mean-field analysis above are borne out; in particular, at large (positive or negative) $\beta, \phi$ we identify the ordered phases described above. We also identify a disordered paramagnetic phase at small $\beta, \phi$.

2. We show that the usual Ising transition persists in a neighbourhood away from $\phi = 0$. For various values of $\phi$, we find a continuous transition and perform a finite-size scaling analysis, finding strong evidence that the lowest critical exponent remains equal to its value for the 3d Ising transition $\nu \approx 0.6299$.

3. However, this transition cannot be driven all the way down to arbitrarily weak string coupling $\phi = -\infty$; as described above, this is precluded by the existence of a new ordered phase corresponding to the nucleation of spherical string worldsheets (either the packed or dilute phase). For the branch-point protocol, we can estimate the smallest value of $g_s$ for which the transition persists by examining the intersection with the $\beta = 0$ line; we find $g_s \approx 0.66$. For the no-touching protocol, we have not attempted to precisely calculate the minimum string coupling, but it is of a similar magnitude.

4. Our technology permits us to measure statistical properties of the topology of domain walls. Intriguingly, we find that the critical point is dominated by *toroidal* domain walls, in that

the average Euler character *per cluster* is zero. This is intriguing, and we do not have a simple explanation for this fact.

In the remainder of this section we present the numerical data that supports the above conclusions.

**Scaling analysis:** In our finite-size scaling analysis we examine the dimensionless *Binder cumulant*, defined as

$$g = \frac{1}{2}\left(3 - \frac{\langle \sigma^4 \rangle}{\langle \sigma^2 \rangle^2}\right). \tag{4.1}$$

Close to a continuous critical point at (say) $\beta = \beta_c$ this dimensionless observable depends on $\beta$ and the system size $L$ through the dimensionless combination $(\beta - \beta_c)L^{\frac{1}{\nu}}$:

$$g(\beta, L) = g\left((\beta - \beta_c)L^{\frac{1}{\nu}}\right), \tag{4.2}$$

where $\nu$ is related to the dimension of the relevant operator that drives the system through criticality. Below we present evidence that the data collapse described by (4.2) occurs with the usual value of the 3d Ising exponent $\nu \approx 0.6299$ even for nonzero $\phi$.

We also examine another observable; given any spin configuration, we study the Euler character divided by the number of clusters $\widehat{N}_C$, i.e.

$$\langle \chi \rangle = \left\langle \frac{1}{\widehat{N}_C}\left(\sum_{\text{faces}} \widehat{W}_\square - \sum_{\text{edges}} \widehat{E}_- + \sum_{\text{vertices}} \widehat{V}_\odot\right)\right\rangle. \tag{4.3}$$

This can be thought of as the average Euler character of a typical domain wall; e.g. deep in the ferromagnetic phase this evaluates to 2 as the typical spin fluctuation results in a small spherical domain wall. We emphasize that this is a non-local observable, since the number of components of the domain wall configuration ($\hat{N}_C$ in the denominator) cannot be determined locally.

**Algorithms used.** We used two algorithms: the standard Metropolis single-spin flip algorithm, and an adaptation of the Wolff cluster algorithm [50] to our modified Ising model. Details of the cluster algorithm can be found in Appendix B. As usual, the single-spin algorithm is useful for mapping the gross structure of the phase diagram away from any phase transitions, and the cluster algorithm is better suited to examining the physics near the critical points. We computed error bars by performing a jacknife analysis on a timeseries that has been binned to remove the effects of autocorrelation; they are extremely small (see e.g. Figure 8), and for the most part we do not show them to reduce clutter. For all scans using the cluster algorithm (i.e. Figures 10 and 13) we verify that errors are stable under binning of the timeseries; for Figure 11 the scan at $\beta = 0$ necessitates that we use the less efficient single-spin algorithm even at the critical point; there close to the critical point the errors (though very small) have not reliably stabilized as a function of binning level. It should be possible to do better by generalizing our cluster algorithm to work at $\beta = 0$; we leave this to later work.

To count the number of components of the domain walls, we implemented a variant of the Hoshen-Kopelman clustering algorithm [51] to the walls themselves. A pitfall of direct application of the Hoshen-Kopelman algorithm is explained in Appendix C.

Our techniques are mostly standard, and we found [52–57] useful in implementing our code.

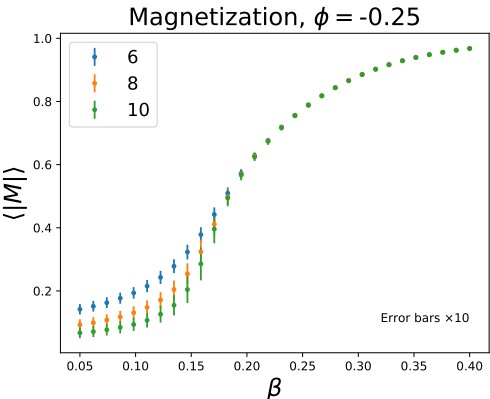

Figure 8: Typical scan of branch point protocol using the cluster algorithm, demonstrating variation of the magnetization for three system sizes; error bars have been scaled upwards by a factor of 10 to make them visible on the plot.

## 4.1 Branch point

We first discuss the overall structure of the phase diagram. In our Monte Carlo simulations, we measure the four order parameters described in Table 1. This is a great deal of information; one convenient way to visualize it is to assign to each point in the $(\beta, \phi)$ plane a color whose RGB values are a function of the four order parameters. In this way we can construct the 2d phase diagram shown in Figure 9; the four ordered phases (FM, AFM, packed, and plumber's nightmare) are clearly visible, as well as a disordered phase at small $\beta, \phi$.

More detailed information can be obtained by performing 1d slices through the phase diagram. We show some illustrative slices varying $\beta$ (at different values of $\phi$) in Figure 10. The slice at $\phi = 0$ is precisely equivalent to the usual Ising model, and $\beta_c \approx 0.22$ as usual. Each slice cuts through the transition between the paramagnetic and ferromagnetic phases; the excellent data collapse for the Binder cumulant shows that the transition remains in the usual 3d Ising class.

We also examine $\langle \chi \rangle$, defined in (4.3). Here we note an extremely curious fact, which is made most obvious by plotting $L^{-3} \langle \chi \rangle$; at the critical point this average Euler character appears to *exactly vanish* for all values of the system size:

$$\langle \chi(\beta = \beta_c) \rangle \to 0 \tag{4.4}$$

as can be seen from the fact that all three curves (with different values of $L$) intersect zero at the same point; the extra factor of $L^{-3}$ helps separate the curves elsewhere. This value for $\beta$ agrees reasonably well with the location of the critical point found by demanding the best data collapse of the Binder cumulant. This suggests that in some sense the average topology of surfaces at the 3d Ising critical point is toroidal. To the best of our knowledge this fact – though a property of

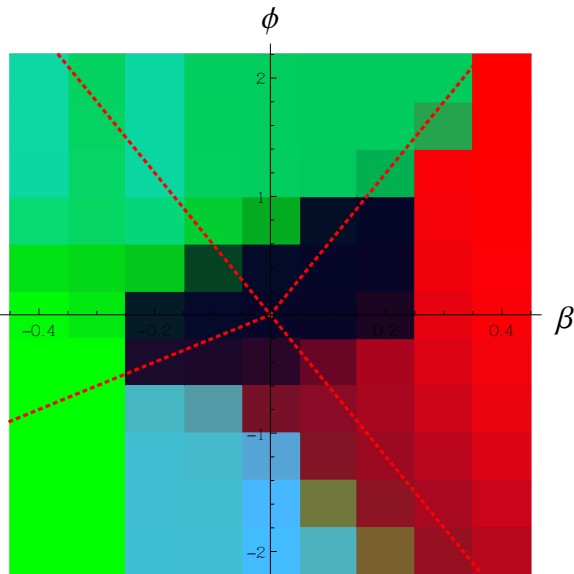

Figure 9: (Color online) Phase diagram on a $10^3$ lattice using the "branch point" vertex resolution protocol. A color coding system has been used where red measures ferromagnetic order, green measures antiferromagnetic order, and blue measures the sum of $O^+$ and $O^-$. The four distinct colors correspond to the following phases in Table 1: top (dark green) is plumber's nightmare I, right (red) is ferromagnetic, bottom (blue) is packed, and left (bright green) is antiferromagnetic. The interior dark region is a paramagnetic phase. The "classical" phase boundaries – i.e. arising purely from minimizing the Hamiltonian – have been indicated with red dotted lines. This is topologically the same as the mean-field phase diagram in Fig. 7 but the phase boundaries have shifted somewhat.

the usual Ising model – has not been observed before, and deserves an explanation in terms of the universal critical Ising field theory[8].

Finally, in Figure 11 we hold $\beta$ fixed at zero and vary $\phi$, finding another path through the transition. Universality dictates that we should now find data collapse with the same exponent as a function of $\phi$ rather than $\beta$, and this is indeed the case. We find $\phi_c \approx 0.415$, corresponding to $g_s \approx 0.66$. In a string theory realization, there is some temptation to take seriously the fact that the bare value of beta (*i.e.* the bare string tension) should be positive; in this case one can

---

[8]Note added: A similar but not identical observation was made by Karowski and Thun in [33] in a model similar to our no-touching model. They studied the average Euler character per unit volume (in contrast to our Figs. 10 and 13, which show the average of the Euler character *per cluster* per unit volume). More extensive simulations by D. Huse [58] then showed that this former quantity does *not* vanish at the critical point. Even if the Euler character per cluster suffers a similar fate, we believe its smallness at the critical point deserves explanation. We thank D. Huse for bringing this work to our attention. An earlier preprint version of this paper on the arXiv also stated that the variance of the average Euler character per cluster was small; further investigation calls this claim into question, and it deserves more study.

view the critical value of $\phi_c$ as being the smallest string coupling at which the familiar 3d Ising transition exists.

## 4.2 No-touching

Results for the no-touching vertex resolution protocol are for the most part similar; a plot of the overall phase diagram shown in Figure 12, and the same 1d slices are shown in Figure 13. Note that the no-touching model at $\phi = 0$ is no longer the pure 3d Ising model; as one might expect, $\beta_c$ shifts from the usual cubic lattice value to $\beta_c \approx 0.178$, but the critical properties appear unchanged. Simulations with this vertex resolution protocol are somewhat more time-consuming, as many potential moves are rejected as they would require touching domain walls.

We note the same curious fact as for the branch point protocol that $\langle \chi \rangle$ appears to vanish at the critical point. The fact that the vertex resolution protocol (and hence the UV regularization of $\chi$) is somewhat different but that $\langle \chi \rangle$ still vanishes at the critical point again hints towards some universal character.

# 5 Discussion

In this work we have presented a local lattice model that endows the usual 3d Ising model with a tunable string coupling. This was performed in an extremely direct manner, by using the fact that the Euler character admits a local integral representation as an alternating sum over vertices, edges, and faces.

Though we have presented some preliminary results, there remains much to explore within this model itself. It would be very interesting to slice open the path integral and study this as a quantum mechanical system in two dimensions as a simple model for stringy dynamics. Perhaps most intriguingly, we found that at the usual 3d Ising critical point the average Euler character of a cluster is zero. This deserves further investigation, both numerical and (optimistically) in terms of the 3d Ising CFT. We note that this observable involves a non-local operator that determines the number of clusters, and it is both challenging and exciting to think about how to access this information[9] from the point of view of the critical theory.

Our primary motivation however was the idea that the Ising transition (or, perhaps, the *approach* to the transition) could be driven to small string coupling, allowing for a perturbative worldsheet description. As the phase diagrams Figures 9 and 12 make clear, this does not quite happen; instead, at small string coupling we find instead a new ordered phase – either the packed or dilute phase – where space is filled with spherical string worldsheets, packed as close together as the UV regularization allows. This is not unexpected: a small $g_s$ penalizes higher genus string worldsheets and thus *encourages* spherical worldsheets, and at sufficiently small $g_s$ this tendency

---

[9]See e.g. [59] for signatures of the 2d Ising critical point in information-theoretic observables.

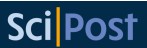

Figure 10: Scans varying $\beta$ while holding $\phi$ fixed at various system sizes, using branch point vertex resolution protocol. *Left column:* Average euler character per domain wall component per unit volume, $\langle \chi \rangle / L^3$. *Right column:* Binder cumulant as a function of scaling variable $(\beta - \beta_c)L^{\frac{1}{\nu}}$. In the Euler number plots, we display a vertical line at $\beta = \beta_c$ (determined from the best collapse of the binder cumulant), and a horizontal line at $\langle \chi \rangle = 0$.

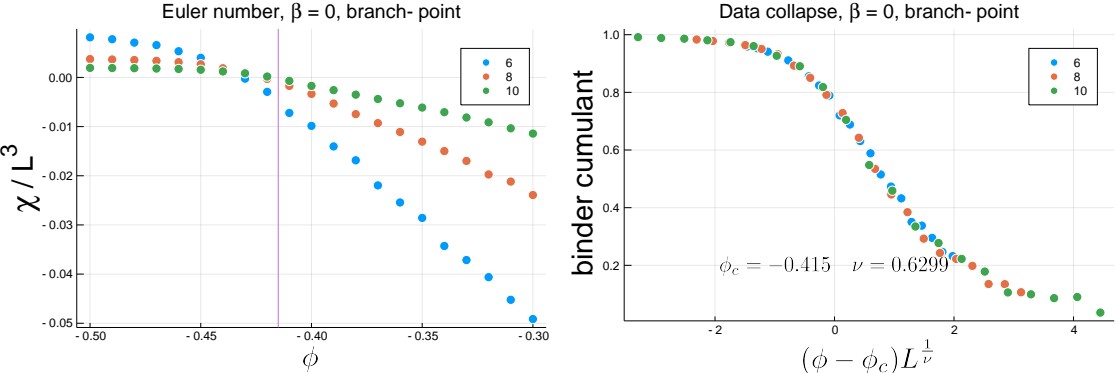

Figure 11: Scans varying $\phi$ while holding $\beta = 0$ at various system sizes, using branch point vertex resolution protocol. *Left column:* Average euler character per domain wall component per unit volume, $\langle \chi \rangle / L^3$. *Right column:* Binder cumulant as a function of scaling variable $(\phi - \phi_c)L^{\frac{1}{\nu}}$.

to nucleate spherical worldsheets overcomes the energy cost associated with their tension.[10] The transition to this new phase is probably first-order, and is not related to the usual Ising transition.

Despite the obstruction of this new string-condensed phase, we still believe that the interpretation of the 3d Ising model as a string theory is an idea that should now be revisited in the light of new technology, both string-theoretical (i.e. holography) and quantum-field-theoretical (i.e. discrete higher-form symmetry). We thus take this opportunity to record some thoughts on this description; at the moment these ideas are still somewhat speculative and far from a quantitative description of the transition, but we anticipate that they form some steps in that direction.

Unlike the remainder of this paper, the following sections assume that the reader has some familiarity with perturbative string theory and holography.

## 5.1 Symmetries on the worldsheet

How does the $\mathbb{Z}_2$ symmetry of the Ising model act in the string theory? We wish to take seriously the idea that the worldsheet is a domain wall between regions of up and down spins.

One point which is not a priori clear is whether we should aim for a dual of the Ising spin model or its gauge theory dual. As discussed in Appendix A, the precise symmetry structure is somewhat different; though the dynamics is the same, different kinds of defect operators are non-local in the two different theories, resulting in the spin model having a 0-form $\mathbb{Z}_2$ symmetry but the gauge theory having a 1-form $\mathbb{Z}_2$ symmetry. It seems that the gauge theory admits more generalizations, in that one may break the 1-form $\mathbb{Z}_2$ symmetry in a controllable fashion by introducing various

---

[10]Framed in this language it is somewhat tempting to interpret the onset of the spherical-worldsheet proliferated phase – a weak-coupling phase where the lowest-energy string mode condenses – in terms of a closed string tachyon condensate; we have however not been able to make this precise in any sense.

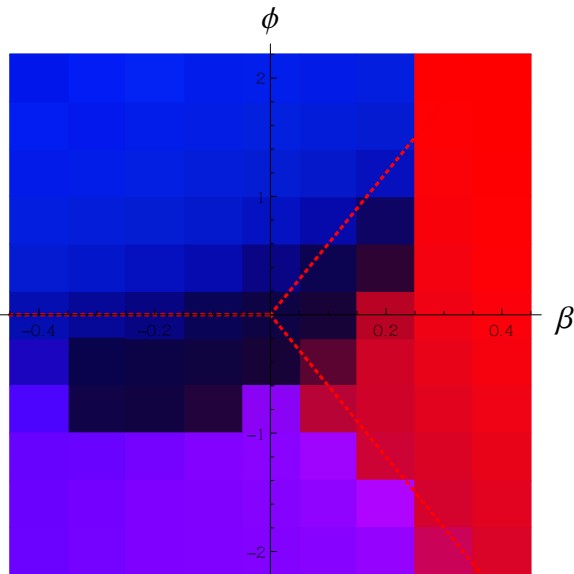

Figure 12: (Color online) Phase diagram on a $8^3$ lattice using the "no-touching" vertex resolution protocol. The same color coding system has been used as for the branch point colorplot in Figure 9. The following phases from Table 1 are realized: top (blue): plumber's nightmare II, right (red): ferromagnetic, and bottom (purple): dilute. The "classical" phase boundaries – i.e. arising purely from minimizing the Hamiltonian – have been indicated with red dotted lines. This is topologically the same as the mean-field phase diagram in Figure 7 but the phase boundaries have shifted somewhat.

kinds of gapped gauge-charged matter. For example, the Ising gauge theory admits a fermion number symmetry, which is carried by the dyon, the boundstate of the $e$-particle (the end of an electric string) and the $m$ particle (the vison).

In either case, the domain walls are unoriented and $\mathbb{Z}_2$-valued – a pair of them can annihilate. This means that the string worldsheet cannot couple to a massless 2-form (NS-NS) gauge field $B_{\mu\nu}$, and should be unoriented. Presumably, we can infer from the fact that $g_s < 0$ [15] that it is an unoriented theory of Sp type rather than O type.

Let us now consider the general form of the spectrum of an RNS superstring upon quotienting by worldsheet parity:

$$
\begin{array}{ccc}
\text{RR} & \text{RR} & e\text{-particle?} \\
\text{NS-R R-NS} \xrightarrow{\text{mod }\Omega} & \text{NS-R} \overset{?}{=} & \text{dyon} \\
\text{NS-NS} & \text{NS-NS} & \text{glueballs}
\end{array}
$$

This theory still has a spacetime fermion number symmetry, which acts by a sign on the NS-R sector. We tentatively identify this sector with the dyon excitation. If we further orbifold by this

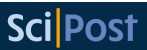

Figure 13: Scans varying $\beta$ while holding $\phi$ fixed for no-touching protocol. *Left column:* Euler character per unit volume $\chi/L^3$. *Right column:* Binder cumulant as a function of scaling variable $(\beta - \beta_c)L^{\frac{1}{\nu}}$. In the Euler number plots, we display a vertical line at $\beta = \beta_c$ (determined from the best collapse of the binder cumulant), and a horizontal line at $\langle \chi \rangle = 0$.

$(-1)^{F_s}$ symmetry, the spectrum takes the form

$$
\text{Orbifold by } (-1)^{F_s}\colon\ 
\begin{matrix} \text{RR} \\ \text{NS-R} \\ \text{NS-NS} \end{matrix}
\ \xrightarrow{\text{mod } (-1)^{F_s}}\ 
\begin{matrix} \text{RR}_L \oplus \text{RR}_R \\ \underline{\quad\quad} \\ \text{NS-NS} \end{matrix}
\ \overset{?}{=}\ 
\begin{matrix} \text{spin} \oplus \text{neutral} \\ \underline{\quad\quad} \\ \text{neutral} \end{matrix}\ .
$$

This (unoriented) type 0 theory now has two RR sectors, labelled by the eigenvalues of the chirality operator $\Gamma$. We conjecture that this global symmetry $\Gamma$ can be identified with the Ising $\mathbb{Z}_2$ symmetry. More work is required to flesh out this interpretation.

## 5.2  Holographic duality

We turn now to dynamics. It is tempting to try to interpret the conjectured duality between the 3d Ising model and a string theory as a holographic duality. Let us now speculate on the structure of the worldsheet description of such a string theory.

As we are (dually) describing a 3d CFT, the string worldsheet must realize the symmetries of the 3d conformal group. One obvious way to accomplish this is to take the target space to be $\text{AdS}_4$:

$$
ds^2 = ds_{AdS}^2 = d\varphi^2 + e^{-2\varphi}\delta_{\mu\nu}dX^\mu dX^\nu , \tag{5.1}
$$

where we denote the putative "holographic" direction with $\varphi$.

We now have an immediate issue. A non-linear sigma model with this (curved) target space is not conformally invariant on the worldsheet; as the target space curvature is *negative*, we find that in the (worldsheet) IR the model flows logarithmically towards $c = 4$ free bosons. This logarithmic flow makes it impossible to define a string theory with this sigma model. We thus need to first find some mechanism to stabilize the RG flow, presumably through a non-trivial zero of the beta function. There is in fact some evidence that such a zero may exist [60], but it involves balancing higher-order corrections against one another and would describe a string-scale target space. (Though unpleasant, this may indeed be what we should expect for the 3d Ising model, which after all has no free parameters that could be used to build a hierarchy between AdS and string scales).

Let us however imagine that somehow a zero of the beta function can be found. The resulting $\text{CFT}_2$ will presumably have $c < 26$, and to construct a critical string theory we must further find a way to cancel the Weyl anomaly. There is some temptation to proceed as in the linear dilaton, and add a term

$$
S_{\text{worldsheet}} \supset S_Q = \frac{Q}{2\pi}\int d^2\sigma\ \varphi(\sigma)R \tag{5.2}
$$

to the worldsheet action, where we have included a linear dilaton field in the $\varphi$ direction, $\phi(\varphi, X^\mu) = Q\varphi$, and where $R$ is the worldsheet Ricci scalar and $Q \sim (c - 26)$ is an appropriately chosen constant. This construction is however incompatible with target space scale invariance, which acts as the AdS isometry $X^\mu \to e^\lambda X^\mu, \varphi \to \varphi + \lambda$. Under this transformation $S_Q$ shifts as

$$
S_Q \to S_Q + Q\lambda\chi . \tag{5.3}
$$

(Indeed, if the dilaton depends on a target-space direction, the theory can hardly be invariant under scaling that direction).

One possible resolution of this puzzle was suggested by Gursoy in [28], in attempts to find a holographic dual of the 3d XY model by realizing an XY transition in a model with a holographic dual. The idea is that a warped AdS can still have spacetime conformal invariance, though it is not manifest in the metric.

Another possible resolution is a 'composite linear dilaton' [61]. The idea is that a composite field $\bar{\varphi}$ can play the role of $\varphi$ in the linear-dilaton coupling:

$$S_Q = \frac{Q}{2\pi} \int d^2\sigma \, \bar{\varphi}(\sigma) R \,, \tag{5.4}$$

where $\bar{\varphi} = \frac{1}{\Delta} \ln \mathcal{O}_\Delta$ is a composite operator which shifts additively under a worldsheet scale transformation. In the case of strings propagating on $AdS_4$, we could choose

$$\mathcal{O}_2 = e^{2\varphi} \partial_\alpha X^\mu \partial^\alpha X_\mu + \partial_\alpha \varphi \partial^\alpha \varphi \tag{5.5}$$

the $AdS_4$ kinetic term (where we assume its dimension $\Delta$ is indeed equal to the canonical value). By construction, this is invariant under target-space scale transformations $X^\mu \to e^\lambda X^\mu$, $\varphi \to \varphi + \lambda$. In this case

$$\bar{\varphi} = \frac{1}{\Delta} \ln \mathcal{O}_\Delta = \varphi + \frac{1}{2} \log \left( |\partial X|^2 + e^{-2\varphi} |\partial \varphi|^2 \right) . \tag{5.6}$$

One can view this as a way of improving the linear dilaton coupling (5.2) to be compatible with *target space* conformal invariance. However, in order for these logarithms to be well-defined, we must focus attention on configurations where the string is extended in some direction.

## 5.3 Effective string theory at criticality

The previous discussion concerned finding a string theory for the entire Ising model. A less ambitious but more concrete connection with string theory does not attempt to model the full partition sum but instead looks for a theory which governs the fluctuations of a large flat domain wall, i.e. the "effective string" [61–63].

In the usual description, worldsheet fields $X(\sigma, \tau)$ fields arise as Goldstone modes for breaking of translations by the wall. 'Large and flat' means $X(\sigma, \tau) = \sigma + $ fluctuations, so $\partial X \neq 0$, and $\log(\partial X)^2$ in the above discussion makes sense. Interestingly, predictions from this theory match lattice simulations [64]. Specifically, they study numerically the ratio of two Wilson loops of the same perimeter but different areas

$$R(L, n) \equiv \frac{\langle W(L+n, L-n) \rangle}{\langle W(L, L) \rangle} e^{-n^2 \sigma} \tag{5.7}$$

close to but not at the critical point; this observable is sensitive to the universal predictions of the effective string theory. It would be interesting to understand the effect of the Euler character term that we have added on the behavior of the Wilson loop, and on its roughening transition.

This discussion has been away from the critical point. Kuti and collaborators [65] find a gapped breathing mode on the worldsheet in this regime. Closer to the critical point, we can expect this mode to become gapless: a Goldstone for breaking of scale transformations by the profile of the wall. This should again be the bulk radial coordinate $\varphi(\sigma, \tau)$. 3d conformal invariance further guarantees that it will assemble together with the Goldstones for translations such that the target space is effectively $AdS_4$ as in (5.1).

We find it striking that for a scale-invariant theory with string excitations, assuming a relation between the effective string theory and a putative more general holographic dual string theory, Goldstone's theorem implies the existence of the radial dimension.

## 5.4 Other assorted puzzles

**Large-$N$ puzzles.** String theory in flat space has Hagedorn growth of single-string states at high energy. In AdS/CFT examples with $N \times N$ matrix-valued fields $X, Y$, this can be matched by the large-$N$ growth of the number of words $\text{tr}(XYXXY\cdots)$ [66, 67]. However our weak-coupling limit does not involve any parameter such as $N$. This is presumably also resolved by a highly-curved target space.

Relatedly, the $g_s \to 0$ limit in string theory is a *classical* limit in the sense that there is a factorization of correlations for the elementary string excitations. In familiar holographic examples, this is realized by large-$N$ factorization. It is not clear to us why or how the dominance of spherical domain walls should imply any such factorization of correlations.

**D-branes?** Would this constitute an unoriented string theory without space-filling D-branes? In all examples we know, RR tadpole cancellation requires D-branes on top of the space-filling O-planes. In contrast, in this construction, the string worldsheets have a $\mathbb{Z}_2$ character and can only end modulo 2. It is possible to introduce external line defects on which the worldsheets can end [64]; possibly such objects would be related to D-branes in some sense, but their dynamics seems like it must be heavily constrained. We note that in the holographic context the idea that string worldsheets can end modulo $N$ can be implemented using a topological term in the bulk action, and is familiar from usual examples of AdS/CFT [68–70]. The discussion of this effect in terms of higher form symmetry in [71] may be useful here.

To conclude: though the set of ideas here is speculative, we hope that they may eventually play a role in a string-theoretical description of critical phenomena.

# Acknowledgements

We thank J. Kuti for discussions on Monte Carlo simulations, E. Fradkin for help with the literature, T. Grover and Y.-Z. You for useful comments, A. Sen and S. Das for discussions about tachyons, and D. Hofman for stimulating discussions about the correctness of Landau. This work

was supported in part by funds provided by the U.S. Department of Energy (D.O.E.) under co-operative research agreement DE-SC0009919, by the Simons Collaboration on Ultra-Quantum Matter, which is a grant from the Simons Foundation (651440, JM), and by the STFC under consolidated grant ST/L000407/1. The numerical calculations in this paper were done using the Julia language.

# A  Symmetries of Ising spin model and gauge theory

In this section, we present a unified discussion of the symmetries and defect operators of the conventional $\mathbb{Z}_2$ Ising spin model on the cubic lattice, as well as those of the closely related $\mathbb{Z}_2$ pure lattice gauge theory. While we frame our discussion in the modern language of higher-form symmetry [7], these considerations are well-known.

First, we review the definition of an Abelian *global symmetry* according to [7]. In a continuum theory defined in $d$ Euclidean dimensions, a $p$-form global symmetry is equivalent to the existence of a conserved charge operator $Q(\mathcal{M})$ which is defined on a $d - (p + 1)$ dimensional closed submanifold $\mathcal{M}$. This charge operator is *topological* in that it does not change under continuous deformations of the surface $\mathcal{M}$. These charges act on charged *objects* $O(C)$, which are defect operators defined on $p$-dimensional surfaces $C$. The statement that $O(C)$ has charge $\alpha$ with respect to the symmetry generated by $Q(\mathcal{M})$ can be phrased as the following operator statement:

$$Q(\mathcal{M})O(C) = e^{i\alpha L(\mathcal{M},C)}O(C), \tag{A.1}$$

where $L(\mathcal{M}, C)$ is the linking number of the two manifolds $\mathcal{M}, C$. As the sum of their dimensions is $d - 1$, such a linking number can always be defined.

The most familiar case is when $p = 0$. Then the $O(C)$ can be thought of as local operators $\mathcal{O}(x)$, and $\mathcal{M}$ is defined on codimension-1 manifolds (e.g. "time-slices"). The fact that $Q(\mathcal{M})$ is invariant under deformations of $\mathcal{M}$ is usually interpreted as the statement that the charge is conserved, and (A.1) is the usual Ward identity for the the local charged operator $\mathcal{O}(x)$. The other case of relevance to us will be $p = 1$; in this case the charged objects $O(C)$ are line operators.

In this section we will work on a cubic lattice; thus we will identify explicit lattice analogues of the above structures.

## A.1  Ising spin model

We first discuss the usual Ising spin model, with Hamiltonian and partition function

$$\widehat{H} = \beta \sum_{\langle ij \rangle}\bigl(1 - \sigma_i \sigma_j\bigr) \qquad Z = \sum_{\{\sigma\}} \exp\bigl(-\widehat{H}\bigr) \tag{A.2}$$

(where as usual in this paper we are absorbing the inverse temperature $\beta$ into the definition of the Hamiltonian).

### A.1.1 Symmetries and charges

This Hamiltonian has a 0-form symmetry for $\mathbb{Z}_2$ spin flips $\sigma_i \to -\sigma_i$. What is the associated charge?

We can identify it by performing a discrete analogue of the usual Noether procedure, where we transform the dynamical fields by a spacetime-dependent symmetry parameter. More specifically, consider a closed 2-dimensional surface $\mathcal{S}$ on the dual lattice, and consider performing the change of variables $\sigma_i \to -\sigma_i$ for all the spins on one side of $\mathcal{S}$. As this is *almost* a symmetry operation, the Hamiltonian remains unchanged away from $\mathcal{S}$, but for every link that crosses $\mathcal{S}$ we find a nonzero contribution:

$$\widehat{H} \to \widehat{H} + \sum_{\langle ij \rangle \in \mathcal{S}} \beta \sigma_i \sigma_j . \tag{A.3}$$

This deformation of the Hamiltonian can be thought of as inserting into the path integral a new *charge operator* $\widehat{Q}_0(\mathcal{S})$ defined on the submanifold $\mathcal{S}$:

$$\widehat{Q}_0(\mathcal{S}) = \exp\left( -\beta \sum_{\langle ij \rangle \in \mathcal{S}} \sigma_i \sigma_j \right) . \tag{A.4}$$

Now consider displacing $\mathcal{S}$ by a few lattice sites; this involves flipping a few more spins $\sigma_i$, those between the old and new locations of $\mathcal{S}$. As this is a change of variables, the partition sum remains unchanged, and thus

$$\sum_{\{\sigma\}} \exp\left(-\widehat{H}\right) \widehat{Q}_0(\mathcal{S} + \delta\mathcal{S}) = \sum_{\{\sigma\}} \exp\left(-\widehat{H}\right) \widehat{Q}_0(\mathcal{S}) . \tag{A.5}$$

In other words, $\widehat{Q}_0(\mathcal{S})$ is topological in the sense described above, and is the desired charge operator. It is also easy to see that the above topological property breaks down if the deformation passes through an explicit spin insertion $\sigma_i$; in other words if we deform $\mathcal{S}$ by a $\delta\mathcal{S}$ such that the point $i$ lies inside $\mathcal{S} \cup \delta\mathcal{S}$ but not in $\mathcal{S}$, we have the following Ward identity[11]

$$\widehat{Q}_0(\mathcal{S} + \delta\mathcal{S})\sigma_i = -\sigma_i \widehat{Q}_0(\mathcal{S}) . \tag{A.6}$$

There is another topological operator that one can define. Consider a curve $C$ connecting sites of the lattice; as we cross each link, we ask if there is a domain wall living on the corresponding site of the dual lattice, and count this number of domain walls modulo 2. In terms of the wall operator defined in (2.4), this is

$$\widehat{Q}_1(C) = \prod_{\square \in C} (-1)^{\widehat{W}_\square} . \tag{A.7}$$

This does not change under small deformations of the curve $C$, and so putatively defines a $\mathbb{Z}_2$ 1-form symmetry. Unlike $\hat{Q}_0(\mathcal{S})$, it is topological "off-shell"; we do not need to sum over the Boltzmann weight (A.5), instead this depends only on the kinematic fact that if the domain wall

---

[11] The ordering of the operators on the right-hand side of (A.6) of course has no meaning, as we are discussing lattice observables in Euclidean classical statistical mechanics; however it is meant to be evocative of the fact that relation descends to a commutator if one slices open the path integral to obtain a Hilbert space interpretation.

configuration is defined in terms of spins all domain walls are closed. This further implies that in the spin formulation of the theory there are no true line operators that are charged under this putative $\mathbb{Z}_2$ 1-form symmetry. If one were being precise, one would say that in this formulation the 1-form $\mathbb{Z}_2$ symmetry does not exist as it has nothing to act on.

### A.1.2  "Topological field theory"

The realization of the symmetry depends on the correlations of the spins $\sigma_i$; if the symmetry is unbroken, then the correlation function $\langle \sigma_i \sigma_j \rangle$ decays exponentially. On the other hand, if the symmetry is spontaneously broken, then we have long-range order with $\langle \sigma_i \rangle \neq 0$.

Let us now consider the extreme infrared of the (gapped) spontaneously broken phase. This theory is almost empty but not quite; note there is now a sense in which the spin operator $\sigma_i$ has also become "topological", as it does not vanish but also has zero correlations with any other non-coincident operator. The only universal information is the symmetry algebra of wrapping operators (A.6), which can be reproduced by a continuum "topological field theory". Generalizing briefly to a $\mathbb{Z}_k$ symmetry, consider the following (almost) trivial theory:

$$S[B, \theta] = \frac{k}{2\pi} \int B \wedge d\theta \,, \tag{A.8}$$

where $B$ is a 2-form and $\theta$ is a scalar. Then we can represent the charge and spin operators as

$$Q_0(\mathcal{S}) = \exp\left( i \int_{\mathcal{S}} B \right) \qquad \sigma(x) = \exp\left( i\theta(x) \right) \,. \tag{A.9}$$

A short calculation reveals that these operators indeed satisfy (A.6) if $k = 2$. We generally do not use this machinery for such a simple problem, but we introduce it here to emphasize that "topological" field theories play a role in *all* spontaneously broken symmetries, whether higher form or conventional.

Note that formally in this spontaneously broken phase we can say that the theory has a microscopic $\mathbb{Z}_2$ 0-form symmetry (whose charge is $Q_0$), and an extra emergent $\mathbb{Z}_2$ 2-form symmetry (whose charge is $\sigma_i = e^{i\theta(x_i)}$, which is topological in this phase).

### A.2  Ising gauge theory

We now discuss the pure $\mathbb{Z}_2$ lattice gauge theory; here the variables $a_l = \pm 1$ are defined on links $l$ of the dual lattice. We construct a field strength defined on faces of the dual lattice:

$$\widehat{F}_{\square}(a) = \prod_{l \in \square} a_l \,, \tag{A.10}$$

where the product runs over the four edges bounding the face. The Hamiltonian and partition sum are simply

$$\widehat{H} = -\tilde{\beta} \sum_{\square} \widehat{F}_{\square}(a) \qquad Z = \sum_{\{a\}} \exp\left( -\widehat{H} \right) \,. \tag{A.11}$$

This model is equivalent under duality to (A.2) [3], where the respective temperatures are related as $\beta = -\frac{1}{2}\log(\tanh\tilde{\beta})$. However the symmetries are realized slightly differently.

### A.2.1 Symmetries and charges

In this formulation, there is a genuine 1-form symmetry. This acts on the dynamical fields as

$$a_l \to \Lambda_l a_l,  \tag{A.12}$$

where $\Lambda_l$ is a $\mathbb{Z}_2$-valued field that lives on the edges, is "closed", i.e. it satisfies the constraint that its product around all faces is 1, i.e. $F(\Lambda)_\square = 1$. This results in a conserved charge: following arguments similar to that above, given a curve $C$ connecting points of the original lattice one can construct the following charge operator, the analogue of (A.7)

$$\widehat{Q}_1(C) \equiv \exp\left(-\tilde{\beta}\sum_{\square \in C} F_\square(a)\right),  \tag{A.13}$$

where the product runs over all the faces intersected by the curve. This is topological in that

$$\sum_{\{a\}}\exp\left(-\widehat{H}\right)\hat{Q}_1(C+\delta C) = \sum_{\{a\}}\exp\left(-\widehat{H}\right)\widehat{Q}_1(C),  \tag{A.14}$$

where to demonstrate this one exploits a $\Lambda_l$ that flips the value of the $a_l$ on links that are intersected by the deformation $\delta C$. The line operator charged under this 1-form symmetry is the Wegner-Wilson line, defined on a curve $C'$ on the dual lattice, i.e.

$$\widehat{A}(C') = \prod_{l \in C'} a_l .  \tag{A.15}$$

This obeys the following Ward identity with the charge operator:

$$\widehat{Q}_1(C)\widehat{A}(C') = (-1)^{L(C,C')}\widehat{A}(C') .  \tag{A.16}$$

What of the 0-form spin flip symmetry? In this formulation this symmetry is obtained even off-shell, i.e. we write

$$\widehat{Q}_0(\mathcal{S}) = \prod_{\square \in \mathcal{S}}(-1)^{F_\square(a)} .  \tag{A.17}$$

Using the decomposition of $F$ into $a$ in (A.10), this clearly depends only on $a$ at the edges of $\mathcal{S}$ and so is topological off-shell. Similarly, there is no local operator that is charged under this symmetry (the spin field $\sigma_i$ is rather non-local in this formalism). If one were being precise one would say that in this formulation the 0-form $\mathbb{Z}_2$ symmetry does not exist as there is nothing for it to act on.

### A.2.2 Topological field theory

We now ask about phases of this model. They should be decided by classifying the behavior of the operator charged under the genuine global symmetry, i.e. the line operator (A.15). If this line operator has an area law behavior, then the 1-form symmetry is *unbroken*; this corresponds to the confined phase of the gauge theory, and the ferromagnetic phase of the spin model. If the line operator has a perimeter law behavior, then the gauge theory is deconfined and the 1-form symmetry is spontaneously broken [7]. Here it is interesting to go to the extreme infrared; there is a sense where the line operator $A(C')$ now has a topological character, as it does not vanish (as it would if it obeyed an area law) but has vanishing correlation with all other non-coincident operators.

Generalizing to $\mathbb{Z}_k$, it is now instructive to define a topological field theory to capture the Ward identity (A.16):

$$S[A_1, A_2] = \frac{k}{2\pi} \int A_1 \wedge dA_2 \, , \tag{A.18}$$

where the charge and Wilson line operators are:

$$\widehat{Q}_1(C) = \exp\left( i \int_C A_1 \right) \qquad \widehat{A}(C') = \exp\left( i \int_{C'} A_2 \right) . \tag{A.19}$$

With the choice $k = 2$, this realizes the algebra (A.16). This is precisely the continuum formulation of $\mathbb{Z}_2$ gauge theory (see e.g. [68,72] for discussion in the high-energy theory literature) and is the description of the "topologically ordered phase". In our language this is equivalent to spontaneous breaking of a 1-form symmetry.

Formally, in this spontaneously broken phase we now have a microscopic $\mathbb{Z}_2$ 1-form symmetry (whose charge is $\widehat{Q}_1$), and an *emergent* $\mathbb{Z}_2$ 1-form symmetry whose charge is $\widehat{A}$.) This emergent symmetry clearly does not exist in the other (unbroken) phase.

To summarize: we see that the 3d Ising spin model has a genuine $\mathbb{Z}_2$ 0-form symmetry that acts on the spin operator, and a $\mathbb{Z}_2$ 1-form symmetry that acts on nothing local and so does not really exist. The pure 3d Ising gauge theory has a genuine $\mathbb{Z}_2$ 1-form symmetry that acts on Wegner-Wilson lines, and a $\mathbb{Z}_2$ 0-form symmetry that acts on nothing local and so does not really exist.

### A.2.3 Charged matter in gauge theory

Though we have discussed the $\mathbb{Z}_2$ gauge theory, we have carefully stayed away from discussing gauge *transformations*; we emphasize that all of our considerations so far have involved only global symmetries. However one often wants to consider adding charged matter to the pure gauge theory; we now describe what this means in our language.

Consider first adding *electric matter*: this is a $\mathbb{Z}_2$ valued field $\phi_a$ defined on sites of the dual lattice. We may now demand invariance under the following gauge transformation:

$$a_{\langle ab \rangle} \to \lambda_a \lambda_b a_{\langle ab \rangle} \qquad \phi_a \to \lambda_a \phi_a \, , \tag{A.20}$$

where $\lambda_a$ is a $\mathbb{Z}_2$ valued *gauge parameter* defined on dual lattice sites. Note that one can now add terms to the action such as $\phi_a a_{ab} \phi_b$ which are invariant under (A.20) but are *not* invariant under the 1-form global symmetry (A.12); thus the addition of electric matter explicitly breaks this symmetry, and the operator (A.13) is no longer topological. The $\phi_a$ form the endpoints of the Wilson line operator (A.15).

We may also add magnetic matter, or *visons*: this is the original spin operator $\sigma_i$. To formulate this in the gauge theory language, we consider acting with the charge operator (A.13) $\widehat{Q_1}(C)$ on a curve with an endpoint $x$; the vison is a single-site defect operator living at the endpoint. The trajectory taken by $C$ away from the endpoint is not important, as it is topological. In the deconfined phase, one can understand the presence of this operator as explicitly breaking the *emergent* $\mathbb{Z}_2$ whose charge is $\widehat{A}(C')$.

# B Cluster updates

Here we describe the adaptation of the Wolff algorithm [50] to our modified Ising model. We follow the discussion of [56].

A cluster is a collection of sites whose spins agree. A cluster is formed, starting with a random site and adding neighboring agreeing sites with a probability $p$. Let $\mathcal{A}(a \to b)$ be a priori probability for constructing in this way a cluster the flipping of which will result in the transition from $a$ to $b$. For a given configuration $a$, let $n_+$ ($n_-$) be the number of bonds across the boundary of the cluster with a $+(-)$ outside. If $a$ ($b$) is the configuration where the spins in the cluster are $+(-)$, then

$$\mathcal{A}(a \to b) = \mathcal{A}_{\text{in}}(1-p)^{n_+}, \quad \mathcal{A}(b \to a) = \mathcal{A}_{\text{in}}(1-p)^{n_-}, \tag{B.1}$$

where $\mathcal{A}_{\text{in}}$ depends on (the size of) the interior of the cluster.

Detailed balance requires

$$\pi(a)\mathcal{A}(a \to b)P(a \to b) \overset{!}{=} \pi(b)\mathcal{A}(b \to a)P(b \to a),$$

where $\pi(a)$ is the Boltzmann weight of configuration $a$, and $P(a \to b)$ is the probability with which we flip the whole cluster. The Boltzmann weight may be written as

$$\pi(a) = \pi_{\text{in}}\pi_{\text{out}}e^{-\beta J(n_- - n_+)}D(a), \tag{B.2}$$

where $\pi_{\text{in/out}}$ contain the dependence on the spins away from the boundary, and $D(a)$ contains the euler character term. $D(a)$ and $D(b)$ only differ at the boundary of the cluster. Therefore the following choice of cluster-flip probability will guarantee detailed balance:

$$P(a \to b) = \min\left(1, \left(\frac{e^{-2\beta}}{1-p}\right)^{n_-}\left(\frac{e^{-2\beta}}{1-p}\right)^{-n_+}\frac{D(b)}{D(a)}\right). \tag{B.3}$$

Making the usual optimal choice of the thus-far-undetermined $p$,

$$p = 1 - e^{-2\beta},$$

gives

$$P(a \to b) = \min\left(1, \frac{D(b)}{D(a)}\right) = \min\left(1, g_s^{\Delta \chi}\right).$$

In the case of $\beta < 0$, (B.3) becomes negative, so we would never add sites to the cluster, and the algorithm devolves to single spin flips. Instead, we can try to build antiferromagnetic clusters. (For $g_s = 1$, the model on a bipartite lattice with $\beta < 0$ is equivalent to the model with $\beta = |\beta|$ by a redefinition of the spins on one sublattice, so in that case, we know that this generalization of the cluster algorithm must work in the case.) That is, form clusters starting from a random site, and adding neighbors which *disagree* with probability $p$. If $a(b)$ is the configuration where the A sublattice is $+(-)$, and $n_+$ $(n_-)$ is the number of bonds across the boundary of the cluster with a $+(-)$ on the A sublattice and $-(+)$ on the B sublattice, then (B.1) remains true, while (B.2) is replaced by

$$\pi(a) = \pi_{\text{in}} \pi_{\text{out}} e^{-|\beta| J (n_- - n_+)} D(a). \tag{B.4}$$

The cluster-flip probability should then be

$$P(a \to b) = \min\left(1, \left(\frac{e^{-2|\beta|}}{1-p}\right)^{n_-} \left(\frac{e^{-2|\beta|}}{1-p}\right)^{-n_+} \frac{D(b)}{D(a)}\right) \tag{B.5}$$

and we should choose

$$p = 1 - e^{-2|\beta|}.$$

## C Comment on clustering algorithm

The Hoshen-Kopelman algorithm [51, 57] is an efficient way to identify connected regions of up spins. It is tempting to try to use this algorithm directly to count the number of connected components of the domain walls separating the regions of up and down spins, $n_{\text{dw}}$. After all, if in a background of down spins we have a $n_\uparrow$ regions of up spins, the number of domain wall components is also $n_\uparrow$. The first problem is that the number of regions of down spins in this situation is only one; flipping all the spins preserves the number of domain walls but changes $n_\uparrow$ to one.

So instead one might try to use $\max(n_\uparrow, n_\downarrow) \stackrel{?}{=} n_{\text{dw}}$. This gives the correct answer for many configurations, including the one on the left of Figure 14. Starting from that configuration (which has $n_\uparrow = 1, n_\downarrow = 2, n_{\text{dw}} = 2$), consider flipping a single spin far from the up spins. This gives $n_\uparrow = 2, n_\downarrow = 2$, but increases the number of domain walls to 3.

For this reason, we instead adapt the Hoshen-Kopelman algorithm to directly count the number of components of the walls themselves. In the case of the branch-point algorithm, this requires keeping careful track of the connections between the walls, as discussed below.

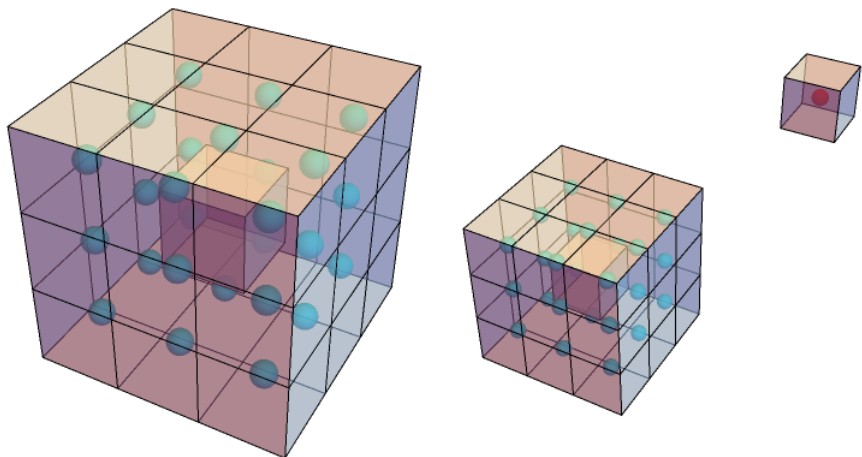

Figure 14: For the left configuration, the number of domains of up spins is $n_\uparrow = 1$, and the number of domains of down spins is $n_\downarrow = 2$; the number of domain wall components is $n_{\mathrm{dw}} = \max(n_\uparrow, n_\downarrow) = 2$. For the right one, this relation fails. Different clusters of up spins are colored differently.

## D  Vertex number assignments

In this section, we explain how the vertex number assignments are constructed. Here "vertex" refers to a vertex of the dual lattice; recall that each vertex is surrounded by eight spins and twelve plaquettes, each of which may be occupied by a domain wall. Each of the $2^8$ spin configurations determines a bit string $\mathcal{V} = \{n_{\square_i}\}$, where for each $i \in 1 \cdots 12$, $n_{\square_i}$ is either 1 or 0 depending on whether a domain wall is present on the corresponding plaquette or not; overall spin flips do not change the domain wall configuration, and thus there are 128 distinct configurations of domain walls.

### D.1  Vertex number

For the purposes of computing the Euler character $\chi$, we must associate a "vertex number" $\mathcal{D}_\mathcal{V}$ to each of the 128 different vertex configurations $\mathcal{V}$. This vertex number counts how many *topologically distinct* surfaces are participating at the vertex, i.e. how many distinct surfaces we get if we "pull apart" the vertex.

To compute the $\mathcal{D}_\mathcal{V}$, we perform the following steps.

1. First, for each vertex configuration $\mathcal{V}^a$, $a \in 1 \cdots 128$, compute the number of domain walls by summing up the bit string:

$$N(\mathcal{V}^a) = \sum_{i=1}^{12} n_{\square_i}^a . \tag{D.1}$$

This number ranges from 3 to 12, with many degeneracies. We sort the $\mathcal{V}^a$ in ascending order by the number of domain walls. Henceforth we assume the $a$ index refers to this sorted list.

2. For each $\mathcal{V}^a$, we ask if the corresponding bit string $n^a_{\square_i}$ can be decomposed in terms of bit strings with *fewer* domain walls, i.e. does there exist a set $\mathcal{S} = \{b_\alpha\}$ of other configs such that for each $b_\alpha \in \mathcal{S}$, we have $b_\alpha < a$ and

$$n^a_{\square_i} = \sum_{b_\alpha \in \mathcal{S}} n^{b_\alpha}_{\square_i}. \tag{D.2}$$

If there is no such $\mathcal{S}$, then we call $\mathcal{V}^a$ a *primitive*; it has $\mathcal{D}_{\mathcal{V}^a} = 1$, and the vertex is touching only a *single* topologically distinct surface.

If such a $\mathcal{S}$ exists, this means that the corresponding $\mathcal{V}^a$ can be pulled apart into simpler surfaces. Typically there will be several such; a further subset $\{\mathcal{S}^I\}$ of these will correspond to *complete* decompositions, in that all the constituent $b_\alpha \in \mathcal{S}^I$ are themselves primitives. Henceforth we will consider only these completely decomposed $\mathcal{S}^I$.

3. For any $\mathcal{V}^a$ there will typically be several completely decomposed $\mathcal{S}^I$, corresponding to the fact that there can be multiple ways to pull apart the vertex into primitives, as shown in Figure 4.

   We must now choose one $\mathcal{S}^*$ to use from this $\mathcal{S}^I$. To make this choice we:

   (a) Pick a decomposition which is rotationally symmetric; i.e. if a subgroup $G$ of the rotation group leaves $\mathcal{V}^a$ invariant, we demand that the chosen decomposition $\mathcal{S}^*$ is also invariant under the same subgroup $G$.

   (b) Given the above constraint, we pick $\mathcal{S}^*$ to have the *largest* number of primitives.

   $\mathcal{D}_{\mathcal{V}^a}$ is then the number of elements in that $\mathcal{S}^*$.

In practice, for all but one configuration one can find a decomposition into the largest number of primitives that is also rotationally invariant. The exception is the maximally symmetric configuration $a = 128$ in the table below, where all 12 plaquettes host domain walls. Here a decomposition into 4 touching cubes is possible but breaks a symmetry; the symmetric decomposition assigns a lower vertex number of 3, corresponding to breaking the configuration into three straight domain walls that intersect at right angles.

The results of implementing this algorithm are shown in Table 2, where we present a picture of the configuration together with the associated $\mathcal{D}_\mathcal{V}$.

Table 2: Table demonstrating results of vertex decomposition algorithm. $a$ runs over all configurations, $\mathcal{V}^a$ displays the pattern of domain walls, and $\mathcal{D}_{\mathcal{V}^a}$ returns the vertex number assigned by the algorithm.

| $a$ | $\mathcal{V}^a$ | $\mathcal{D}_{\mathcal{V}^a}$ | $a$ | $\mathcal{V}^a$ | $\mathcal{D}_{\mathcal{V}^a}$ | $a$ | $\mathcal{V}^a$ | $\mathcal{D}_{\mathcal{V}^a}$ |
|---|---|---|---|---|---|---|---|---|
| 1 | | 0 | 2 | | 1 | 3 | | 1 |
| 4 | | 1 | 5 | | 1 | 6 | | 1 |
| 7 | | 1 | 8 | | 1 | 9 | | 1 |
| 10 | | 1 | 11 | | 1 | 12 | | 1 |
| 13 | | 1 | 14 | | 1 | 15 | | 1 |
| 16 | | 1 | 17 | | 1 | 18 | | 1 |
| 19 | | 1 | 20 | | 1 | 21 | | 1 |
| 22 | | 1 | 23 | | 1 | 24 | | 1 |
| 25 | | 1 | 26 | | 1 | 27 | | 1 |
| 28 | | 1 | 29 | | 1 | 30 | | 1 |
| 31 | | 1 | 32 | | 1 | 33 | | 1 |
| 34 | | 1 | 35 | | 1 | 36 | | 1 |
| 37 | | 1 | 38 | | 1 | 39 | | 1 |

| 40 | 1 | 41 | 1 | 42 | 1 |
|---|---|---|---|---|---|
| 43 | 1 | 44 | 1 | 45 | 1 |
| 46 | 1 | 47 | 1 | 48 | 1 |
| 49 | 2 | 50 | 2 | 51 | 2 |
| 52 | 1 | 53 | 2 | 54 | 1 |
| 55 | 2 | 56 | 1 | 57 | 1 |
| 58 | 2 | 59 | 2 | 60 | 1 |
| 61 | 1 | 62 | 1 | 63 | 2 |
| 64 | 1 | 65 | 1 | 66 | 2 |
| 67 | 2 | 68 | 1 | 69 | 1 |
| 70 | 1 | 71 | 2 | 72 | 1 |
| 73 | 1 | 74 | 1 | 75 | 1 |
| 76 | 2 | 77 | 2 | 78 | 2 |
| 79 | 2 | 80 | 2 | 81 | 2 |

| 82 | | 2 | 83 | | 2 | 84 | | 2 |
|----|---|---|----|---|---|----|---|---|
| 85 | | 2 | 86 | | 2 | 87 | | 2 |
| 88 | | 2 | 89 | | 2 | 90 | | 2 |
| 91 | | 2 | 92 | | 2 | 93 | | 2 |
| 94 | | 2 | 95 | | 2 | 96 | | 2 |
| 97 | | 2 | 98 | | 2 | 99 | | 2 |
| 100 | | 2 | 101 | | 2 | 102 | | 2 |
| 103 | | 2 | 104 | | 2 | 105 | | 2 |
| 106 | | 2 | 107 | | 2 | 108 | | 2 |
| 109 | | 2 | 110 | | 2 | 111 | | 2 |
| 112 | | 2 | 113 | | 2 | 114 | | 2 |
| 115 | | 2 | 116 | | 2 | 117 | | 2 |
| 118 | | 2 | 119 | | 2 | 120 | | 3 |
| 121 | | 3 | 122 | | 3 | 123 | | 3 |

| 124 | | 3 | 125 | | 3 | 126 | | 3 |
|---|---|---|---|---|---|---|---|---|
| 127 | | 3 | 128 | | 3 | | | |

## D.2 Branch points

As explained in Section 2.2.4, it is sometimes required to introduce a *branch point* along an edge; this happens when the respective decompositions (i.e. the choices of $\mathcal{S}$ above) of the two vertices bounding the edge *disagree*, in that the walls on either side of the edge are reconnected in incompatible ways. To be more precise, this reconnection data is stored in the set of primitive $b_\alpha$ making up each $\mathcal{S}$ – we say that two walls are *connected* if they both belong to the *same* $b_\alpha$ in $\mathcal{S}$. (This is compatible with the idea of "pulling apart" the vertex into the $b_\alpha$.) We must now check which patterns of wall reconnections result in conflicts.

In practice, given a vertex we consider the 6 edges coming out of the vertex; we call the part of the edge associated with the vertex a "nubbin". (Clearly each edge has two nubbins, one from each vertex bounding it). For each nubbin, we enumerate the four walls incident on the corresponding edge as in Figure 1. We now store a number for each nubbin, corresponding to the following possibilities:

1. Only a single domain wall; in this case the two decompositions of the vertices across the edge are guaranteed to agree.

2. Two domain walls, connected as 1-2, 3-4.

3. Two domain walls, connected as 1-3, 2-4.

4. Two domain walls, connected as 1-4, 2-3.

Each of these "nubbin numbers" is stored in a $128 \times 6$ lookup table, indexed by the same $a$ as the vertex lookup table above. When we evaluate the Euler character of a configuration, we consider for each edge the nubbin numbers of its two nubbins. If the two nubbin numbers disagree, then we must introduce a branch point along the corresponding edge: as explained in Section 2.2.4, the resulting curvature singularity means that we count the edge as 3 rather than as 2.

From the point of view of the spin Hamiltonian, this can be thought of as adding a particular vertex-vertex interaction as in (2.16).

## E   Arts and Crafts

Follow the instructions below to make your own branch point, as discussed in §2.2.4. When correctly assembled the eight gray arcs (each contributing an angle of $\frac{\pi}{2}$) should form one continuous curve, illustrating that the full angle about the branch point is $4\pi$.

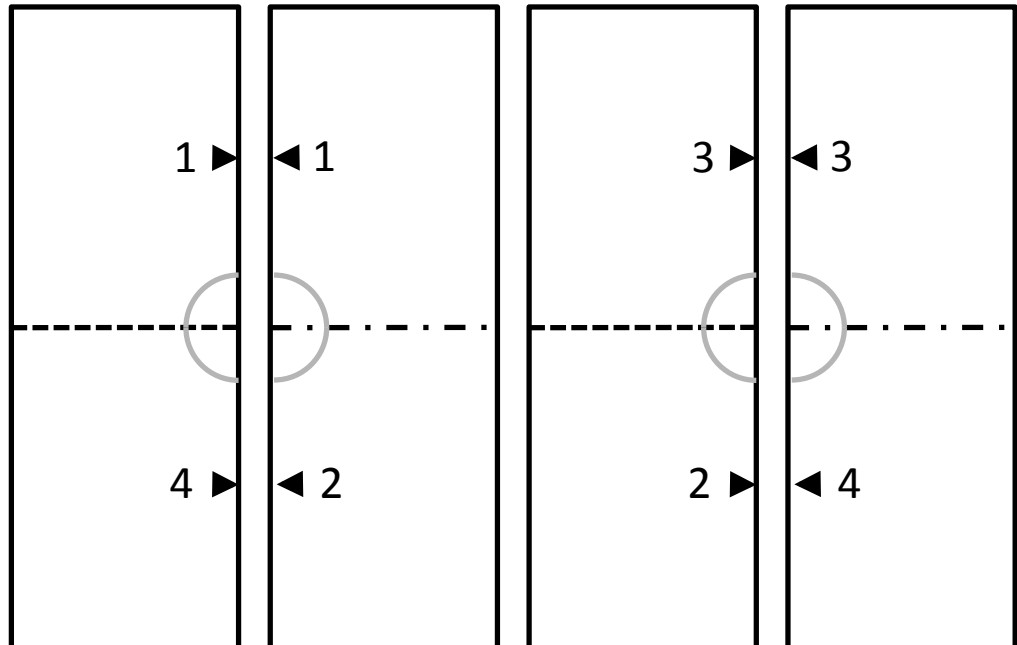

Cut out along solid black lines.
Fold ------ towards you and — · — · — away from you.
Tape together two parallel edges 1 – 1, 2 – 2, etc.

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
