# Peer review of "Toward a 3d Ising model with a weakly-coupled string theory dual"

_SciPost Physics, doi:SciPost Phys. 9, 019 (2020)_

## Round 2 · Referee Report · Anonymous (Referee 1) · 2020-7-1

Report

It has long been conjectured that the 3D Ising model is dual to some sort of string theory, where the domain walls of the ordered phase are supposed to be the image(s) of the string worldsheet. A precise formulation has, alas, proven elusive. The string theory is almost certainly strongly-coupled (|g_s|=1) and there's the delicate matter of how to count the configurations where the string is immersed, rather than embedded.

There are two well-known formulations for dealing with this issue: either one imposes self-avoidance (which is computationally-expensive and not quite right, because domain wall are allowed to touch) or one takes g_s=-1 and hope that contributions from immersions of non-oriented string worldsheets cancel the "bad" configurations.

The authors have embarked on a rather nice Monte-Carlo study of these two alternative formulations, with variable string coupling, hoping to map out the phase structure of the string theory.

This is an interesting, and highly worthwhile avenue to pursue. I recommend this paper for publication.

---

## Editorial Decision

published